# Biomimetic fibrous semiconducting micromesh via tuning phase separation for high-performance stretchable optoelectronic synapses

Qing Zhou [1], Xinzhao Xu[1], Gezhou Zhu[1], Wenhao Li[1], Haoqing Zhang[1], Lin Shao[1], Zhihui Wang [2] ✉, Yunqi Liu[1] ✉ & Yan Zhao [1,2] ✉

Polymer semiconductors hold great potential for next-generation bionic devices, due to their inherent flexibility and biocompatibility. However, endowing them both robust mechanical properties and significant functionalities remains challenging. Bioinspired microstructures can effectively boost semiconducting properties and functionality, yet the structure engineering strategy in conjugated polymers (CPs) systems is underdeveloped. Here, we fabricate biomimetic hybrid semiconducting films featuring geometry-deformable micromesh and nanofibril substructure, through the Van der Waals force-mediated phase-separation. Poly(butyleneadipate-co-terephthalate) (PBAT), an aggregating polymer with abundant intermolecular interactions, is employed as plastic component to facilitate the formation of hierarchically biomimetic structure. Consequently, this geometry-deformable micromesh and interpenetrating phases significantly enhance mechanical and electrical stretchability of the semiconductors. The dependence of strain dissipation mechanism on structural parameters is identified for micromesh structure optimization. Moreover, the nanofibril substructure significantly improves photosensitivity by 100%. Leveraging the synergistic effect of micromesh and nanofibril, synaptic phototransistors are fabricated, which exhibit superior synaptic plasticity and robust performance under strains up to 125% and 1000 repeated cycles at 50% strain, well imitating the phototransduction and memory functionalities of visual system. This strategy shows great potential for processing ultra-stretchable and high-performance conjugated polymer films aiming at stretchable bioelectronics.

Conjugated polymers-based organic field effect transistors (OFETs) are promising candidates for the development of embodied intelligence[1–6], for the merit of inherent mechanical compliance[7,8], potential for high mobility[9] and biophysically analogous properties to the human brain[10–15]. Among them, OFET-based optoelectronic synapses are gaining attention for the potential to emulate visual perception, adaptation, and imaging[5,16,17]. In particular, the development of photo-responsive synapses combining high stretchability will boost advancements in

[1]Laboratory of Molecular Materials and Devices, Department of Materials Science, Fudan University, Shanghai, China. [2]Department of Respiratory and Critical Care Medicine, Changhai Hospital, Naval Medical University, Shanghai, China. ✉e-mail: zhw_ecust@163.com; liuyq@fudan.edu.cn; zhaoy@fudan.edu.cn

many fields. Despite its promises, most of the reported synaptic OFETs exhibited degradable performance under deformation, due to the lack of conjugated polymers (CP) with high optoelectronic performance and stretchability[10,11,13,18,19].

To achieve functional and mechanical properties comparable to those of biological tissues, biomimetic structure design provides a comprehensive strategy[20–22]. In nature, many unique and efficient functionalities stem from structural features such as (1) the porous architectures, (2) hierarchical structure, and (3) gradient composition. Inspired by these principles, we propose that both stretchability and functionality of CP films can be enhanced by engineering their morphology into hierarchical structure that mimics biological tissues, which possesses the targeted functionality.

To impart stretchability into CP films, bio-inspired film morphology engineering—particularly through geometrically deformable structures—has gained traction[23–25]. Geometric deformation in these structures, e.g., mesh structure and porous structure, allows the film to accommodate the tensile strain, thereby preserving the crystallinity of semiconducting layer under large strain. For example, honeycomb-like CP films fabricated using template-based approaches have demonstrated enhanced stretchability and improved interactions with electrolytes, enabling sensitive electrocardiogram (ECG) signal detection under repeated strains up to 30%[24]. However, such template-based methods typically require polar solvents (e.g., water) for demolding, which can adversely affect charge transport in n-type polymer semiconductors. Alternatively, a more versatile and efficient method is to incorporate elastomer into CP to form an interpenetrating network comprising CP-rich phase and elastomer-rich phase[23,25]. In this architecture, CP-rich domains maintain efficient charge transport pathways, while elastomer-rich domains dissipate mechanical stress. Notably, this strategy is capable to modulate the film morphology into structures resembling biological tissues, e.g., mesh structure[25] and fibrous textures[26,27], offering potential for enhancing bioinspired functionalities. However, current studies have yet to integrate these structural features to achieve optimal performance. Moreover, significant challenges remain, particularly in mitigating the effects of pronounced phase separation, which can disrupt the continuity of both charge and strain transport, leading to degradation under large deformation. Furthermore, current methods overlook the need for improved functionalities, such as photosensitivity. These challenges underscore the urgent need for rational design strategies in the hybrid material systems, as current research has mainly focused on the stretchability of elastomer, while ignored other factors that determine the film morphology.

In this work, we present a facile and versatile methodology for fabricating ultra-stretchable CP films by tuning phase separation dynamics through Van der Waals force interactions. This approach yields a biomimetic film characterized by a fibrous micromesh (FMM) structure with a record-low characteristic feature size. The FMM structure imparts high electrical stretchability to the polymer semiconductors and leads to a two-fold enhancement in photoresponsivity in the Thieno[3,2-b]thiophene-diketopyrrolopyrrole (DPPTT)-based FMM film. To optimize the photoelectrical performance and stretchability, we investigated the relationship between the strain dissipation and the pore diameter. Our experimental and theoretical analyses revealed a nonlinear increase in stretchability with pore diameter, identifying a critical threshold beyond which strain dissipation mechanism changes. Leveraging the FMM structure, we fabricated stretchable optoelectronic synapses that mimic visual memory functions. The devices demonstrate stable and robust synaptic behaviors under repeated large strains, paving the way for their application in next-generation sensing and neuromorphic computing platforms.

## Results and discussion

### Tailoring stretchable hybrid film with a biomimetic fibrous micromesh structure

In biological vision system, light sensing and conversion into neural signals are primarily executed in retina, specifically the photoreceptor cells, which feature abundant fiber-like structure arrangement favorable for light-detecting efficiency. To mimic phototransduction capacities of visual system and mechanical compatibility with natural tissue, high photosensitivity and stretchability are desired for semiconducting polymers to construct phototransistors. Based on the concept of function-oriented biomimetic structure design, a highly stretchable hybrid semiconducting polymer film with an FMM structure was fabricated via precisely tuning phase separation (Fig. 1a). This approach aims to enhance interpenetrating network of semiconducting phase and plastic phase and facilitate the fibrous structure by fast chain assembly during casting. To start with, the solution mixture was prepared by dissolving the organic semiconductor (OSC) and a stretchable insulator Poly (butylene adipate-co-terephthalate) (PBAT) in the chloroform. After fabrication, the interpenetrating network of OSC and PBAT is formed in the hybrid film, where the continuous OSC provides charge transport path and PBAT serves as the stretchable component under deformation. Poly{[N,N9-bis(2-octyldodecyl)-naphthalene-1,4,5,8-bis(dicarboximide)−2,6-diyl]-alt-5,59-(2,29-bisthiophene)} (N2200) was chosen as the semiconducting phase for its high electron mobility and representative semicrystalline microstructure[28]. Chemical structures of PBAT and N2200 are listed in Fig. 1a. In order to accelerate the chain assembly during phase separation, PBAT with intensive intermolecular Van der Waals force was adopted as the plastic component (Fig. 1a). Differing from commonly used elastic insulators which exhibit amorphous structure, PBAT possesses high stretchability based on a semicrystalline structure (Supplementary Fig. 1), in which polar ester group in the backbone and the hydrogen bond contained in the terminal groups provide strong driving force for self-assembly during casting. The intensive intermolecular interactions were confirmed by the high surface energy. As shown in Supplementary Fig. 2 and Supplementary Table 1, PBAT exhibited a high surface energy of 52.3 mJ m$^{-2}$, much greater than the commonly used elastomer Styrene-Ethylene-Butylene-Styrene (SEBS). The semicrystalline structure and well-aggregated fibril texture of PBAT were characterized by the grazing incidence wide-angle X-ray scattering (GIWAXS) and atomic force microscope (AFM), respectively (Supplementary Fig. 3 and Fig. 4). Meanwhile, the repeated alkyl segments in PBAT play a role of flexible hinges between the aggregated chains, endowing it great stretchability.

After one-step spin coating, a hierarchically assembled film with micrometer mesh and a substructure of retina-like fibrils was obtained from the prepared hybrid solution. To make a clear insight into the hybrid film with FMM structure, AFM was performed at different length scales, as shown in Fig. 1b. Micromesh structure was composed of interpenetrating semiconductor-rich and isolator-rich regions. Both phases featured fibrous substructure with diameters in tens of nanometers, which resembled the pristine PBAT and N2200 film morphology, respectively (Supplementary Fig. 4 and Fig. 6). This enhanced aggregation in the hybrid film was also confirmed by the GIWAXS and UV–visible absorption spectrum (Supplementary Fig. 5, Fig. 7 and Fig. 8). Compared to the neat film, FMM film exhibited a stronger absorption at the 0−0 peak near 709 nm and a weaker absorption at the 0−1 peak near 640 nm, indicating stronger chains aggregation in FMM film. Based on these results, Fig. 1c illustrates the hierarchically ordered structure of FMM film from the aggregates to the macroscopic scale. Owning to the intensive Van der Waals force, N2200 and PBAT aggregates interpenetrate with each other and assemble into the macroscopic FMM.

The influence of the component proportion on the hierarchical morphology was further investigated. The proportion of PBAT was

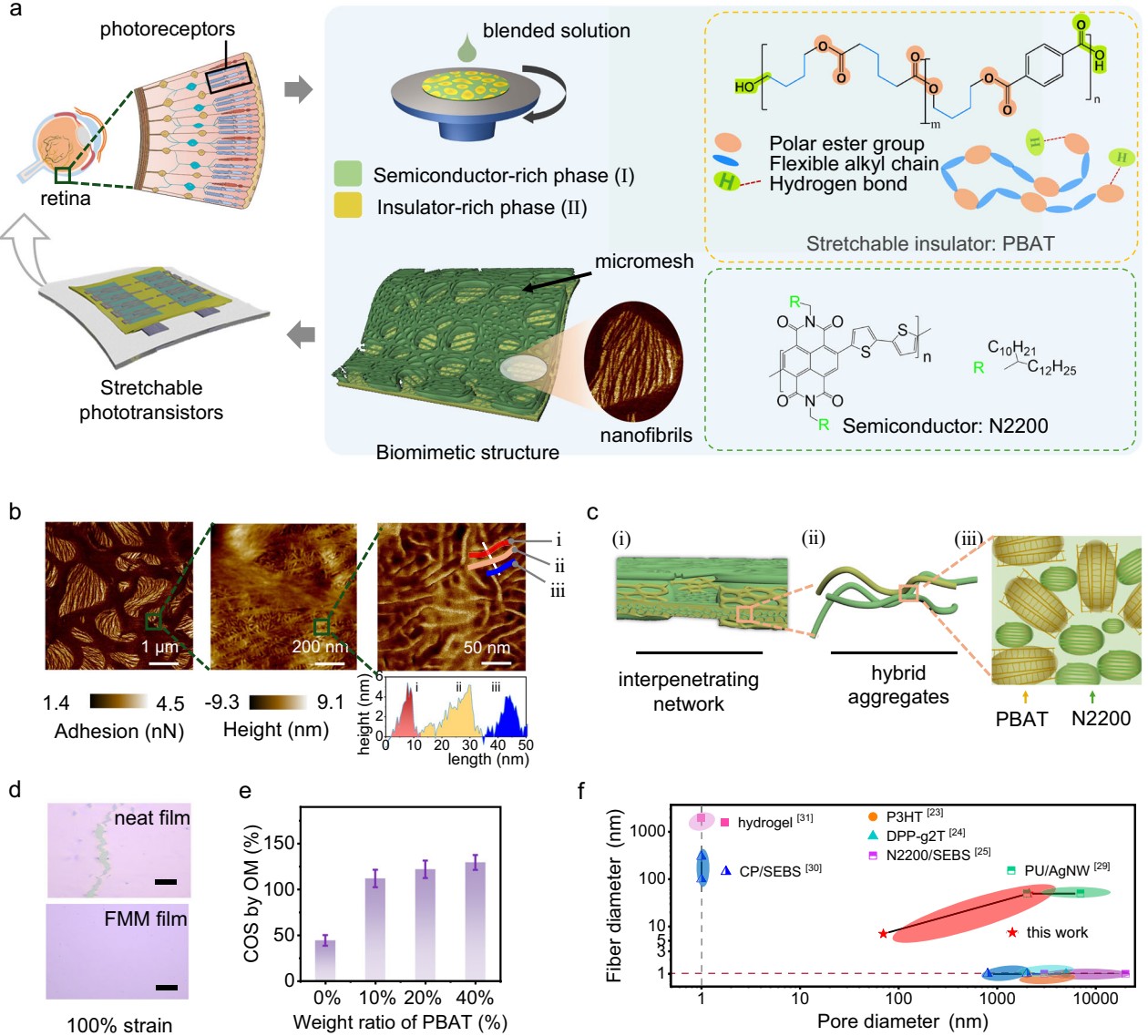

**Fig. 1 | Fabrication of FMM film of stretchable semiconducting polymer through incorporating semicrystalline plastic polymer. a** Biomimetic design of organic semiconductors for stretchable bioelectronics (left). Schematic of FMM film preparation through spin-coating method and chemical structure of the N2200 and PBAT (right). **b** AFM images of FMM film at different length scales. To distinguish the phase separation, adhesion (left) and height (middle and right) are shown. To demonstrate the fibrous texture of FMM films, a line cut profile across three fibers (i, ii, iii) is shown at the bottom right. **c** Illustration of the hierarchical ordered structure of FMM film. **d** Optical images of neat N2200 film (upper) and FMM film (below) under 100% strain. Scale bar is 20 µm. **e** Crack-onset strain (COS)

of FMM films with different component contents observed by optical microscope (OM). Data are presented as mean values ± SD, $n = 5$. Data were obtained from five independent parallel tests. Detailed morphology of each film is presented in Supplementary Fig. 9. **f** Comparison chart by plotting the fibril diameter (unit: nm) versus pore diameter (unit: nm) among stretchable films fabricated through various strategies (i.e., P3HT[23], DPP-g2T[24], N2200-SEBS[25], PU/AgNW[29], CP/SEBS[30], hydrogel[31]). Points under the red dash line represent films with no fibrils. Points at the left side of the gray dash line represent films with no porous/mesh structures. See the list of Abbreviations for full chemical names.

controlled within the range of 10% to 40%, and the corresponding morphology of hybrid films under different post-treatment conditions were investigated by topography- and adhesion atomic force microscope (AFM) (Supplementary Tables 2, 3). With the increasing ratio of insulator from 10% to 40%, the dimension of PBAT-rich phase (region with lower height and higher adhesion) became larger (ranging from hundreds of nanometers to micrometers). Notably, as the proportion of PBAT increased to 40%, the fibrous texture became more prominent, in which nanofibrils bridge across the micro pores and construct intertwined networks. In addition, post-annealing treatment has negligible impact on the film morphology.

Furthermore, the relationship between mechanical stretchability and component proportion was studied through quantifying crack-

onset strain (COS) as the characteristic parameters (Fig. 1d, e and Supplementary Fig. 9). The COS test was conducted on the PDMS substrate with a thin layer of SEBS. Comparing to the neat N2200 film with COS of 50%, the introduction of PBAT enhanced the COS of hybrid film, highest to 130% (Fig. 1f). The fundamental electrical performance of hybrid films with different PBAT contents was characterized (Supplementary Figs. 10, 11). Due to the intrinsic insulation of PBAT, increasing content led to decreased carrier mobilities, as detailed electronic parameters summarized in Supplementary Table 4. But the interpenetrating texture preserved the effective charge transport pathways even in high content of insulator up to 40%. PBAT imparts stretchability into the hybrid films in two ways: (1) dissipating strain energy through chain deformation throughout the film and (2)

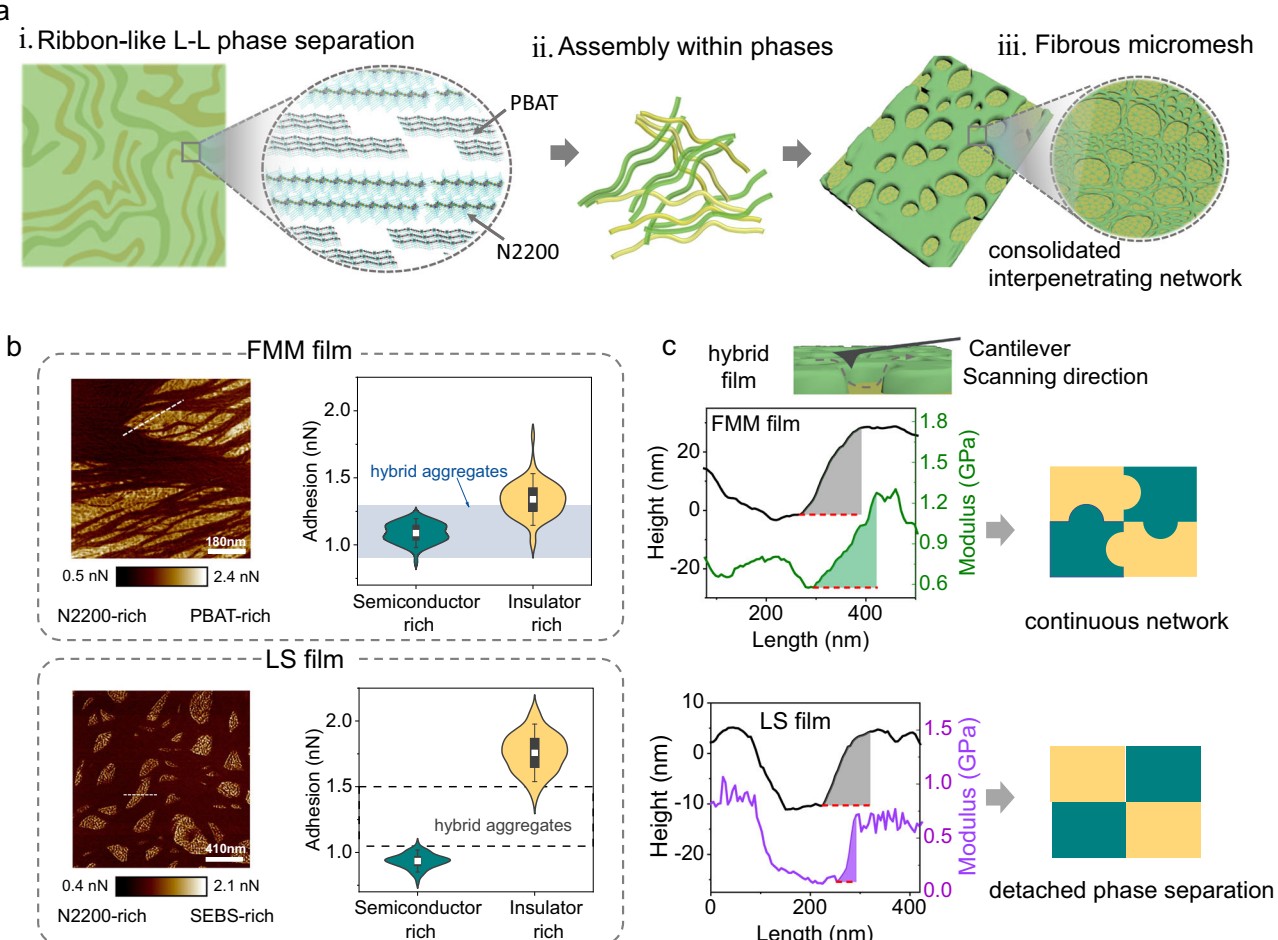

**Fig. 2 | Interpenetrating network in the fibrous micromesh via Van der Waals force tuning phase separation. a** Consecutive schematics of phase separation with intensive aggregation. (i) L-L separation; (ii) aggregation-assisted coarsening and (iii) the resultant fibrous micromesh. **b** (left) Adhesion AFM images of the elastomer-rich and semiconductor-rich regions in the FMM- and LS films. The scale bar is 180 nm and 410 nm, respectively. Both films were fabricated using spin-coating method with the speed of 1500 rpm. The corresponding topography AFM images were detailed in Supplementary Fig. 13. (right) Statistical profiles of DMT modulus extracted from the adhesion AFM revealed the continuous network and macro phase separation in FMM film and LS film, respectively. The filled/unfilled boxes illustrate adhesion distribution of hybrid aggregates. **c** (left) Detailed schematic of the AFM mapping in the cross-section region and the tracks of height and modulus. See the list of Abbreviations for full chemical names. (right) Illustration of the semiconductors and insulator distribution within the film. The green panel represents the N2200, the yellow panel represents the insulator.

tailoring the film morphology towards a geometrical deformable structure. Considering the balanced device performance and extraordinary film morphology, hybrid film with the insulator weight ratio of 40% was chosen for further applications in optoelectronics.

The above results have demonstrated that this strategy contributes to fabricating biomimetic polymer films with tuneable pore diameter and exquisite nanofibers, significantly improving the mechanical stretchability of PBAT/N2200 hybrid film. Notably, the substructure of nano fibrils in our FMM film exhibit a record-small size among materials incorporated in stretchable electronics (Fig. 1f)[23–25,29–31].

**Van der Waals force tuning phase separation**

To understand the formation of hierarchical fibrous mesh structure, phase separation coupled with the intensive chain assembly is considered (Fig. 2a). As depicted in Supplementary Fig. 12, during the spin coating process, phase separation undergoes two sequential kinetics phrases: liquid-liquid (L-L) phase separation and subsequent coarsening[32,33]. Theoretical investigations suggest that subtle concentration fluctuations in the liquid mixture initiate spatial L-L phase separation. During this phase, the compositions of the coexisting phases are contiguous, and the phase length scales are minimal. As

solvent evaporation progresses, molecular chains gradually congregate, leading to the formation of organic semiconductor-rich and plastic-rich domains, which results in large phase domains and distinct compositions (coarsening stage). With the evaporation of the solvent, the diffusion of the molecular is more hindered, and the coarsening stage will be terminated after the solvent is completely volatilized. The resultant film features dual-phase network, which is highly determined by the molecular behavior in both phases.

The FMM film features with micromesh morphology constructed from nanoscale fibrils, which is dramatically different from typical cylinder sphere or lamellar structure. The intensive intermolecular interactions in PBAT play an important role in the formation of this fibrous mesh structure, via tunning the competition between chain assembly and chain diffusion. As the mixed solution concentrates, short-range interactions (Van der Waals force) induce PBAT chains to form clusters. Such localized condensation leads to minor L-L phase separation. Owing to the inherent rigidity, PBAT chains tend to exhibit long persisting length, which extends the molecular cluster along backbone direction, resulting in ribbon-like L-L separation at nanoscale. This structure is transferred into the film morphology via chains assembly after coarsening phase. The formation of ribbon-like L-L separation facilitates the growth of

interpenetrating hybrid aggregates, which prevents further phase separation via chain diffusion. In consequence, FMM is formed through this aggregation-assisted quenching. Due to the "quenching" effect of aggregation, the progression of composition heterogeneity and the enlargement of the phase region are both hindered, resulting in an interpenetrating network and smaller length scale in the resultant dry films.

To elucidate the influence of Van der Waals force on the phase separation in hybrid films, a comparative study was conducted using hybrid systems incorporating elastomers with different intermolecular interactions. As a counterpart to PBAT, SEBS, which features typical amorphous structure without any crystallinity, was selected to fabricate another hybrid film, lateral phase separation(LS) film. The interpenetrating network of CP and plastic insulators was analysed along both surface and depth profiles. The weight ratio of plastic components was maintained at 40% in both systems, consistent with our prior findings. To begin with, the typical length scale was studied via scanning electron microscopy (SEM). FMM film exhibited a smaller length scale (averaged ~2 μm) than the LS film (averaged ~5 μm) under identical processing conditions as shown in Supplementary Fig. 24.

Subsequently, statistical analysis of AFM was utilized to highlight the distinct phase separation in the two types of films. In AFM observation, OSC and elastomers were discernible due to their contrasting adhesion and modulus properties (Supplementary Fig. 14). OSC-rich regions exhibited higher Derjaguin–Muller–Toporov (DMT) modulus and low adhesion, while elastomers utilized in this study exhibited lower DMT modulus and higher adhesion. The measured DMT modulus of the pristine N2200 and PBAT are $798 \pm 81$ MPa and $358 \pm 39$ MPa, respectively (Supplementary Fig. 14). Notably, SEBS film exhibited two distinct DMT modulus, with $127 \pm 19$ MPa in the soft segment and $1.35 \pm 0.23$ GPa in the rigid segment, attributable to its copolymeric architecture comprising both rigid and soft segments. According to the statistical results in Fig. 2b, LS film showed clearly separated adhesion distributions in OSC- and elastomer-rich domains, both close to that of pristine films. In contrast, adhesion distributions in FMM films exhibited pronounced overlap across the coexisting phases. These results indicated a more continuous network in FMM film. Similar phasic distribution was observed in the DMT modulus distributions (Supplementary Fig. 15). For instance, while similar height differences were observed in both films near the cross-section, FMM film exhibited a wider interpenetrating layer in modulus profile (Fig. 2c). Furthermore, the interpenetrating network in FMM film along the depth direction was confirmed via elemental analysis by X-ray photoelectron spectroscopy (Supplementary Fig. 16). These distinct phase separation in the FMM- and LS films underscores the influence of molecular interactions on the morphology of hybrid films. In summary, tuning phase separation by screening hybrid components based on intermolecular force has proven to be an effective strategy to achieve highly interpenetrating network in the hybrid CP films, which potentially enhances mechanical properties and functionalities.

### Investigation on the dependence of strain dissipation on pore diameter

The mechanical properties of porous films can be strongly influenced by the pore diameter. Thus, it is essential to understand the relationship between the strain dissipation capacity and pore diameter of FMM structure. The pore diameter can be adjusted by controlling the ratio of components and the processing parameters. A series of hybrid solutions with the insulator component proportions from 10% to 40% were prepared and fabricated into films using spin-coating method at 1500 and 3500 rpm, as well as drop-coating method on a 75° tilted substrate (Fig. 3a). In general, pore diameters increase with the increasing proportion of insulators, which can be modulated from hundreds of nanometers to micrometers. Meanwhile, higher dissolvent speed refines the pore diameters effectively. Consequently,

corresponding porous hybrid films with characteristic pore diameters varying from 175 nm to 2100 nm were obtained as shown in Fig. 3b, c.

In stretched porous films, the strain energy can be dissipated by internal chain deformation and external geometry deformation. Strain-induced chains alignment can provide a comprehensive understanding of the stress within the polymer chains. Subsequently, the polarized ultraviolet–visible absorption spectroscopy (polarized UV–vis) was employed to probe the chain alignment and the geometric deformation in stretched porous films (Fig. 3d). The anisotropy of polymer chains can be quantified by the dichroic ratio ($R$), i.e., the ratio of peak absorbance parallel and perpendicular to the polarization direction. The dichroic ratio of hybrid films increases with strain over the studied range, indicating stress induces chain alignment, as shown in Supplementary Fig. 17.

To investigate the influence of pore diameter on stress dissipation under strain, the orientation parameter

$$f = (R - 1)/(R + 1) \tag{1}$$

was plotted versus the pore diameter (Fig. 3f). As the pore diameter increased from 175 nm to 2100 nm, $f$ decreased from 0.22 to 0.1 within a threshold zone (260 nm to 700 nm). This change in $f$ indicates a shift in the primary mechanism of energy dissipation with varying pore diameter. When the diameter is smaller than 260 nm, strain energy predominantly accumulates within the polymer chains. Conversely, when the diameter exceeds 700 nm, geometric deformation undertakes primary functions for energy dissipation, thereby preserving the chains from significant conformational changes. Thus, to enhance strain tolerance, it is essential to employ a relatively large pore diameter, specifically larger 700 nm. For stretchable semiconducting films, the relationship between strain dissipation and geometric dimensions of porous structure have not been previously elucidated, despite its significance for practical application. Here, by correlating the strain-induced alignment with pore diameter, we reveal a nonlinear relationship in which pore diameter that facilitates stretchability, with a critical zone identified as the threshold. These results deepen our understanding of the strain energy dissipation mechanism in porous fibril structure.

Guided by experimental results, we further investigated the stress distribution in the stretched film featuring varying pore diameters. Finite element analysis was performed to simulate the behavior of free-standing porous films under uniaxial tensile stress (Supplementary Note 1). Two model films with pore diameters differing by a factor of ten were subjected to uniaxial tensile stress, as depicted in Fig. 3f. Typical thickness of CP films fabricated via spin-coating ranges from 50 to 100 nm. Considering the pore diameter corresponding to different energy dissipation regimes (170 nm at chains deformation region and 2000 nm at geometry deformation region), pore diameter was set to be $3 d$ (for model P1) and $30 d$ (for model P2) to represent conjugated porous films in the respective regions, where $d$ is the thickness of both model films. When subjected to an absolute tensile strain of 25%, both the bridge and island regions in model P1 exhibited relatively high stress concentrations. In contrast, for model P2, stress levels in the island and edge regions decreased by over one-third compared to those in P1 model. This indicates that a greater proportion of the strain was dissipated through the geometric deformation of the pore structure (Fig. 3g). The distinct stress distributions in the two model films were consistent with previously obtained UV–vis results. High stress concentration aligns polymer chains to higher degree of orientation and leads to compromised mechanical reversibility. In summary, the correlation between pore diameter and polymer chain alignment provides reference for processing CP films to balance biological and mechanical requirements.

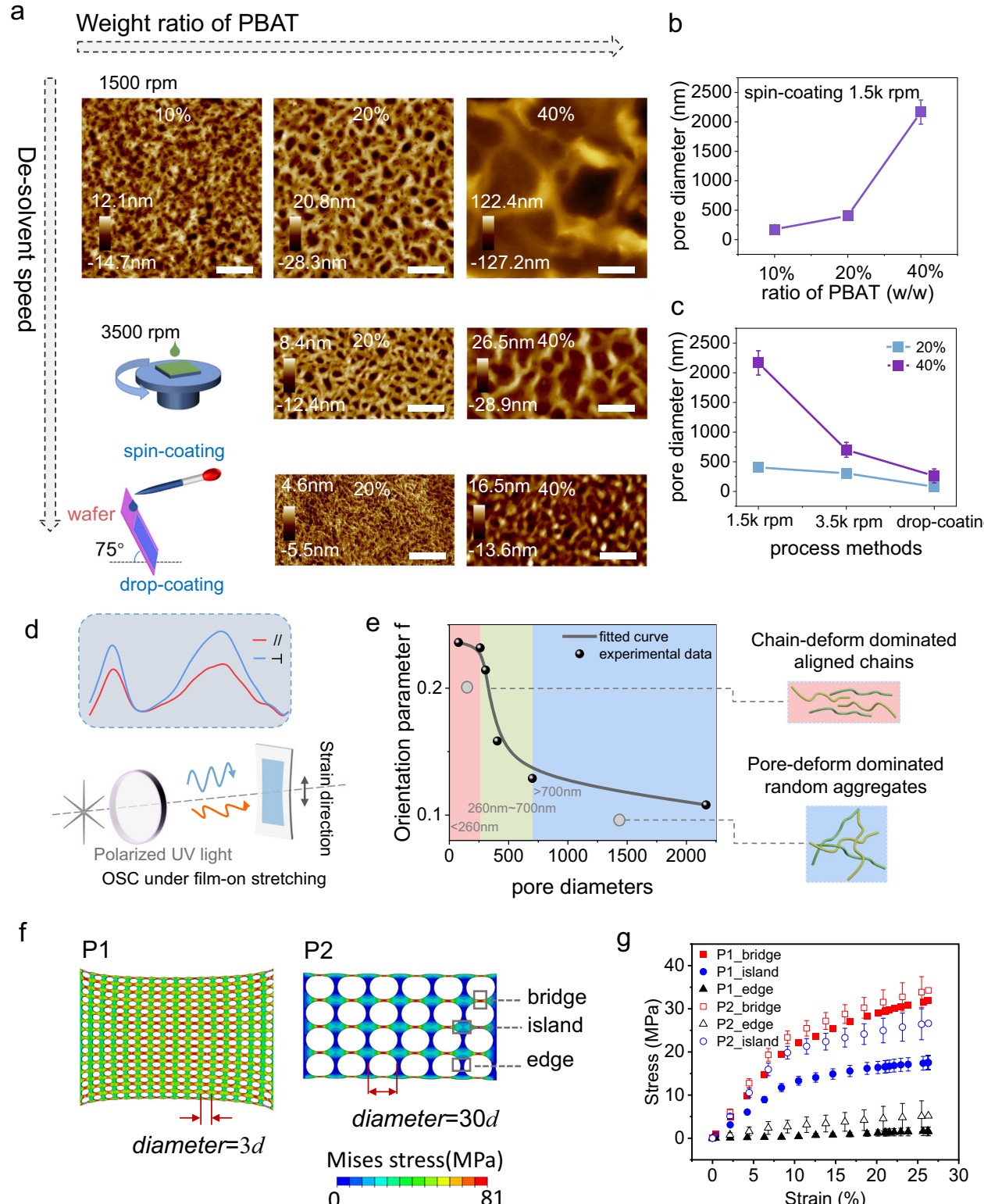

**Fig. 3 | Controlling film geometry towards strain-resistant morphology.**
**a** Topography AFM of hybrid films obtained under different process methods. The scale bar is 1 μm. **b** The characteristic pore diameters in the porous OSC films fabricated from spin-coating methods (1500 rpm) by varying proportion of PBAT. Data are presented as mean values ± SD, $n = 10$. **c** The characteristic pore diameters in the porous OSC films fabricated from various methods. Data are presented as mean values ± SD, $n = 10$. **d** Schematic of the polarized UV−vis absorption spectroscopy for OSC polymer films under various strains. **e** The orientation parameter f of hybrid porous films as a function of the pore diameters. The pink panel represents films with pore diameters smaller than 260 nm. The green panel represents films with pore diameters larger than 260 nm but smaller than 700 nm. The blue panel represents films with pore diameters larger than 700 nm. The corresponding polarized UV−vis absorption spectrum is detailed in Supplementary Figs. 18 and 19. **f** Finite element prediction of stress distribution in porous films with different pore diameters. *d* represents the thickness of the OSC films. Both models were initialized with the same film thickness. **g** Stress distribution in the bridge, island and edge regions in the modeling films upon strain. Data are presented as mean values ± SD, $n = 3$.

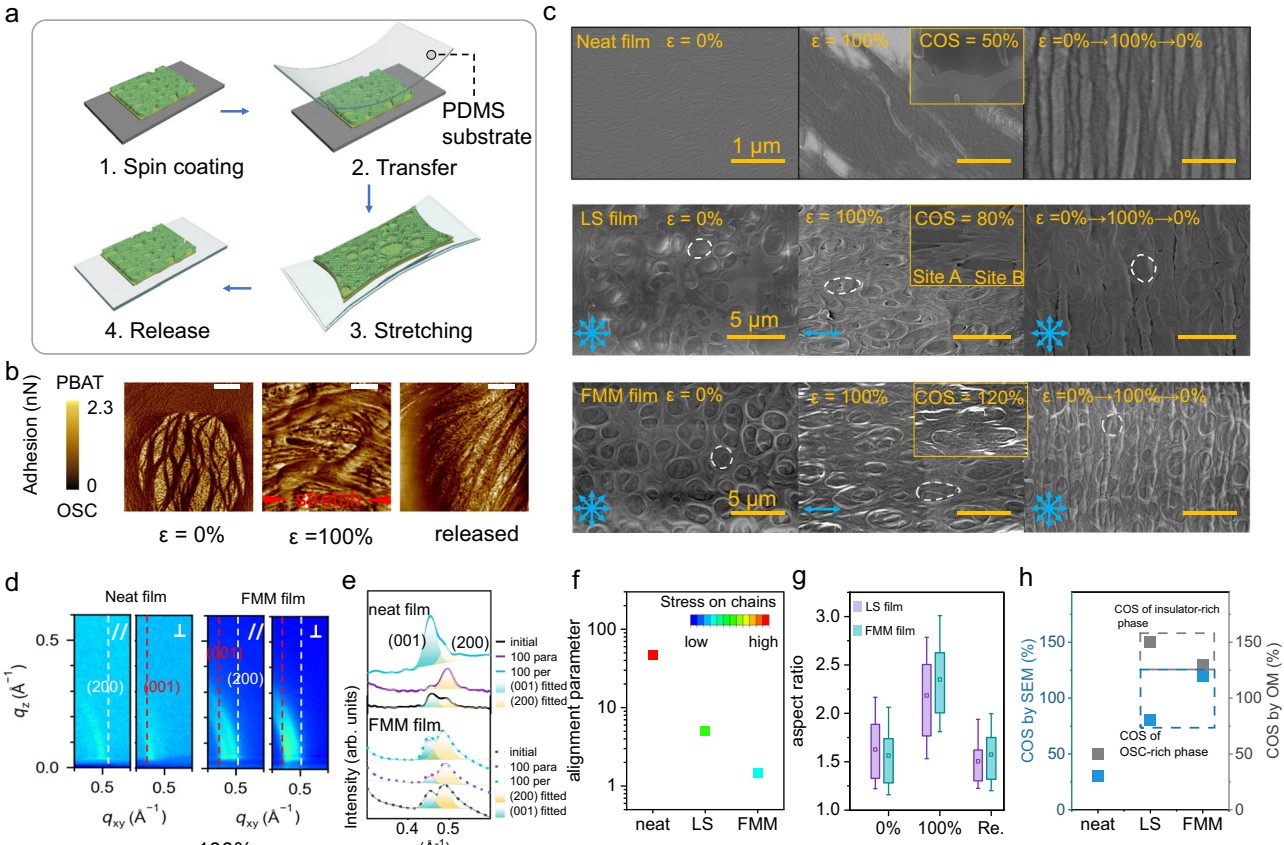

**Fig. 4 | Film morphology of the investigated films under varying stretching conditions. a** Illustration of the film-on-elastomer stretching strategy. **b** Adhesion AFM images of FMM films at various strain (ε represents strain). Scale bar is 200 μm. **c** SEM images of the neat film, LS film and FMM film at initial state, under 100% strain and released from 100% strain. Insert images demonstrate the morphology of OSC films when cracks initiate. Site A: the OSC-rich region; Site B: the insulator-rich region. **d** GIWAXS patterns of the neat films and FMM films of N2200 under 100% strain. "//" and "⊥" represent the incident light parallel and perpendicular to the stretching direction, respectively. The white and red line mark the signal (200) and (001), respectively. **e** Profile of the GIWAXS patterns of the neat film and FMM films at initial state and under 100% strain, integrated of the azimuth angle ranging from 0° to 180°. **f** Alignment parameter in neat film, LS film and FMM film under 100% strain. Alignment parameter is defined as the ratio of the integral intensities of the (001) peak when measured with incident light parallel and perpendicular to the strain direction. **g** Aspect ratio of the pores of LS and FMM films captured by SEM. **h** COS for neat film, LS film and FMM film observed by SEM and OM. See the list of Abbreviations for full names.

## Mechanical properties of FMM film

In porous stretchable films, multiple energy dissipation mechanisms, including geometric deformation and chains deformation, synergically contribute to the strain resistance. Note that the predominant geometric deformation effectively protects the chains from damage, thereby potentially achieving high mechanical reversibility. Besides, the interpenetrating network provides continuous pathway for deformation, avoiding extreme concentration of stress in phase interface and delaying fatal fracture under strain. Here, the improvement of the stretchability of FMM film was investigated across multiple scales. For all the characterizations, FMM films were transferred onto a stretchable substrate and applied to varying strains (Fig. 4a)[34,35]. Firstly, the morphological response of the fine-tuned nanofibrils to tensile strain was investigated using AFM. FMM films under investigation here were fabricated by spin-coating hybrid solution with a PBAT proportion of 40% at 1500 rpm. As shown in Fig. 4b, upon stretched to 100% strain, these suspended nanoribbons exhibited significant alignment along the tensile direction without any cracks or breaks observed, which was attributed to the efficient strain energy dissipation through deformation of hierarchically porous structure, minimizing the strain applied to polymer chains. Once strain was released, these nanoribbons quickly reverted to fibrous structure.

To further elucidate the structure deformation process, SEM was utilized to observe the film morphology under varying strain. For comparative study, the neat N2200 film and LS film were also investigated. For clarity, the structural difference of neat film, LS film and FMM films is summarized in Supplementary Table 5. As shown in Fig. 4c, significant pore deformation under tensile strain was observed in both FMM- and LS films, conforming to remarkably improved COS, 80% of LS (Fig. 4c, LS film, insert (site A)) and 120% of FMM film. Meanwhile, the severe wrinkled morphology in neat film after releasing strain, indicating irreversible deformation, was greatly alleviated by porous structures, especially in FMM film. Although PBAT film showed less resilience than SEBS film, FMM films based on PBAT component still exhibited high reversibility comparable to LS film when stretched to same strain level (Supplementary Figs. 20 and 21).

The well-preserved microstructure in the OSC films was also confirmed by the GIWAXS characterization (Fig. 4d, Supplementary Fig. 22). In CP films, diffraction peaks of (200) and (001) represent the existence of lamella packing (normal to the chains) and repeated backbones along the chains, respectively[35]. Upon stretching to 100% tensile strain, the neat N2200 films exhibited pronounced anisotropic diffraction for both peaks, with (200) peak intensifying in the direction parallel to stretching direction and the (001) peak in the direction perpendicular to stretching, indicating significant strain-induced chain alignment. In contrast, FMM films maintained stable diffraction intensities for both (200) and (001) peaks even when stretched to double strain. To visualize the alignment of crystalline region in neat-,

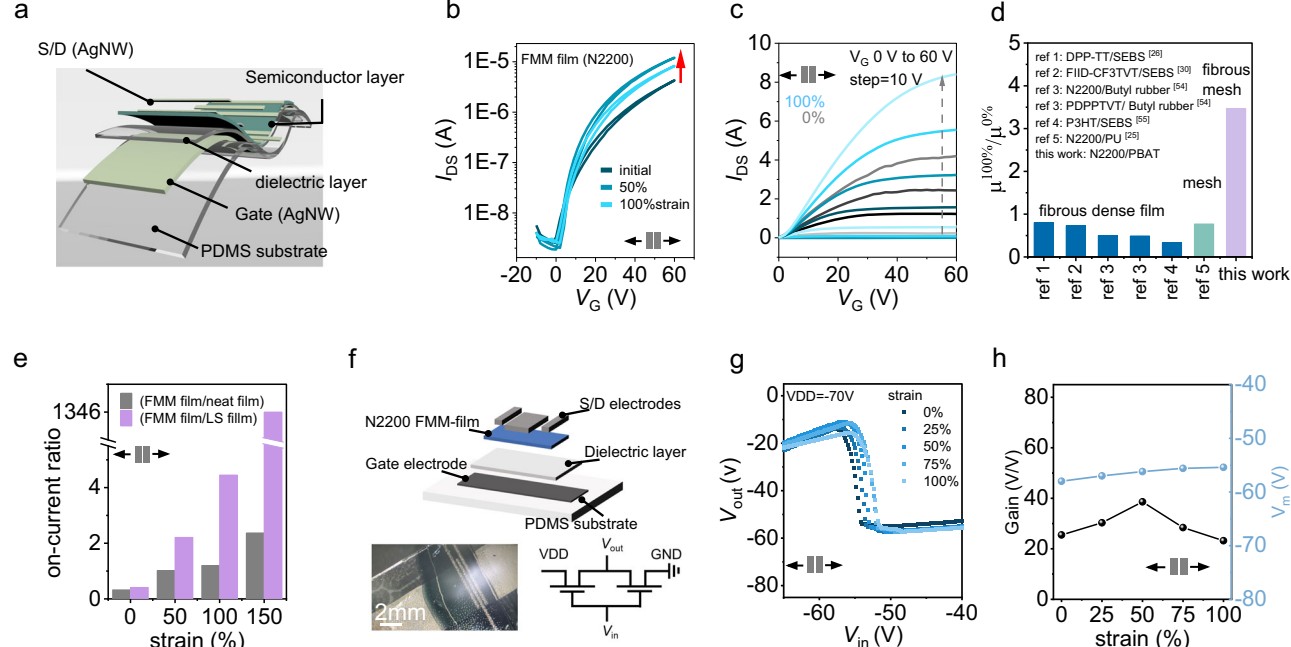

**Fig. 5 | Intrinsically stretchable organic transistors and inverters employing FMM films of N2200. a** Device structure of the stretchable OFET. The initial channel length and width were 100 μm and 5000 μm, respectively. Channel length and width, and the capacity of the stretchable dielectric layer under biaxial strain are listed in the Supplementary Table 5. **b** Transfer curves of the fully stretchable OFETs employing FMM film, neat film and LS film of N2200 ($\varepsilon$ = 0%, 50% and 100%). **c** Output curves at initial state and 100% strain of the OFET in (**b**). **d** Comparison of mobility retention of stretchable OFETs employing conjugated polymer/elastomer blends in reported works and this work, i.e. DPP-TT/SEBS[26], FIID-CF3TVT/SEBS[30], N2200/Butyl rubber[54], PDPPTVT/ Butyl rubber[54], P3HT/SEBS[55], N2200/PU[25], N2200/ PBAT. All the OFETs were stretched to 100% strain. **e** Comparison of the on-current of the stretchable OFETs employing neat film, FMM film and LS film of N2200 under various strain. The on-current was extracted from the transfer curves at $V_G$ = 60 V, $V_{DS}$ = 60 V. **f** Schematic of the stretchable organic complementary inverter with the optical microscope image and the circuit diagram. **g** Representative voltage transfer curve (VTC) of the stretchable inverter under various strain along the charge transport direction. **h** Parameter changes of device performance under various strains along the charge transport direction.

LS- and FMM- film under 100% strain, the ratio of integral of peak (001) with strain parallel to and perpendicular to the radiation incident direction is plotted in Fig. 4f. Upon stretching to 100% strain, the alignment parameter in the neat film increases to 47 due to the tension applied to the chains. Whereas the alignment parameter in the LS and FMM films increases by less than 10. This stability is attributed to the geometric deformation that effectively countered the applied strain, thereby preserving crystalline morphology within FMM films.

Furthermore, a statistical analysis of the porous structure in FMM- and LS- film was implemented to figure out the impact of the interpenetrating network on the film morphology under tensile stress. Due to the distinct elastomeric phase, FMM film exhibited a smaller

characteristic length of pore structure compared to LS film via same fabricating process conditions (Supplementary Fig. 23). When stretched to 100% strain, the pores in FMM films exhibited a higher average aspect ratio compared to those in LS films, demonstrating the capability of interpenetrating network to relieve stress concentration. This film morphology also enhanced crack resistance in semiconductor-rich region by reducing stress concentration at this area (Fig. 4c, Supplementary Figs. 20 and 21). FMM film exhibited slightly lower COS than LS films from observation of optical microscope, where the macro cracks observed under 50-fold magnification were presumed to occur in the insulator-rich region (Supplementary Figs. 25–27). This phenomenon is generally anticipated as the PBAT is less stretchable than the SEBS (Supplementary Fig. 1). Yet, the higher strain resistance in the OSC-rich region endows FMM film better electrical stretchability. To conclude, multi-dimensional observation and comparison have confirmed that the deformable micromesh and the interpenetrating network significantly improve the mechanical stretchability of FMM films, especially at large tensile strain.

**Stretchable organic field effect transistors (OFETs) and inverters**
FMM film was employed as active layer to fabricate stretchable organic transistors with bottom-gate-top-contact structure for further investigation on its electrical properties. Figure 5a illustrates the device structure of the fully stretchable transistors. For comparison, the electrical properties of stretchable transistors based on neat N2200 film and LS film were also investigated. Transfer curves and output characteristics under varying strains were measured to evaluate the electrical performance (Fig. 5b, c, Supplementary Figs. 28–31). Detailed parameters of the stretchable organic transistors based on the neat, LS and FMM film were extracted and summarized in Table 1. At initial state, all the transistors exhibit typical performance with negligible

#### Table 1 | Parameters of the fully stretchable organic transistors

| Materials Contents | Device state | $\mu_{sat}$ (cm² V⁻¹ s⁻¹) | $V_{th}$ (V) | $I_{off}$ (nA) | $I_{on}$ (μA) | Current retention (%) |
|---|---|---|---|---|---|---|
| Neat film | initial | 0.16 | 0.26 | 18.9 | 13.3 | 55 |
| | 100% strain | 0.28 | 10.6 | 9.68 | 7.35 | |
| FMM film | initial | 0.08 | 0.34 | 3.03 | 3.87 | 228 |
| | 100% strain | 0.33 | 2.2 | 1.49 | 8.84 | |
| LS film | initial | 0.21 | 6.5 | 51.5 | 10.5 | 11 |
| | 100% strain | 0.075 | 13.9 | 50.1 | 1.16 | |

$\mu_{sat}$ represents the field effect mobility extracted from the saturation region; $V_{th}$ represents the threshold voltage; $I_{on}$ was extracted from the transfer curves at $V_G$ = 60 V, $V_{DS}$ = 60 V; $I_{off}$ was extracted from the transfer curves at $V_G$ = 0 V, $V_{DS}$ = 60 V. Current retention is the ratio of $I_{on}$ under 100% strain to $I_{on}$ at initial state. Mobility is calculated with the consideration of devices geometry change and dielectric capacitance (Supplementary Table 6).

hysteresis. Compared to the neat film and LS film, the FMM-film based transistor demonstrated a higher current on/off ratio ($I_{on}/I_{off}$) of $1.26 \times 10^3$ and a low threshold voltage of 0.34 V, though a minimal decline of mobility from 0.16 $cm^2 V^{-1} s^{-1}$ to 0.08 $cm^2 V^{-1} s^{-1}$ is observed. At stretched state, FMM film showed stable performance even at large strains. When stretched to 100% strain, N2200-based FMM film exhibited a high current retention of 200% that of the initial state, significantly outperforming LS film, neat film or other hybrid films with porous structure or fibrous structure (Supplementary Figs. 28–31, Fig. 5d, Table 1). Note that FMM film-based transistors maintained typical functions even at 150% strain, attributing to the effective structure deformation and stress dissipation mechanism of hierarchically FMM; whereas stretchable OFETs based on neat N2200 film and LS film showed slight decrease of $I_{on}/I_{off}$ under large strain (>50%), mainly caused by the cracks formed in OSC regions (Supplementary Figs. 30, 31). Notably, due to severe lateral phase separation in the LS film, poor cohesion between the insulator-rich phase and the semiconductor-rich phase led to contact failure under large strain. In general, FMM film exhibits much more stable and robust device performance under various strain, highlighting the improvement of the FMM structure on the electrical stretchability (Fig. 5e, Table 1). Meanwhile, we also investigated the stability of film morphology and device performance under different temperatures and humidity. As shown in Supplementary Fig. 32, the fibrous mesh structure of FMM films of N2200 exhibited minimal changes after being stored at nitrogen or ambient conditions for 1 month. Devices on $Si/SiO_2$ substrates exhibited stable performance, where high mobility retention of 83% and 59% was obtained in the devices stored in nitrogen and humidity control cabinet, respectively. The stretchable OFETs also demonstrated stable performance under a strain of 50% after 30 day storage in nitrogen and ambient conditions (46% RH).

To further validate the functional stability in practical applications under tensile strain, FMM film was employed in stretchable complementary inverters (CT-inverters) as a vital building block of logic circuits, which consist of one p-type transistor and an n-type transistor[36–38]. Here, we fabricated stretchable CT-inverters with one active layer of FMM film, taking advantage of the bipolar charge transport in N2200. The optical microscope image and schematics of the devices are shown in Fig. 5f, respectively. The detailed fabrication process is described in the Supplementary information. Figure 5g shows the output curves of a representative stretchable CT-inverter. As the input voltage swept from −70 V to 0 V, the output voltage switched from a high level to a low level correctly, with a gain of 25.5. The stretchable CT-inverter operates stably during stretched along the channel direction, with only minor changes in the switching voltage observed (Fig. 5g). Consistent with the strain-enhanced electrical performance of the transistors, the stretchable CT-inverters exhibit a minor increment of Gain at 50% strain (from 25.5 to 38.6). These results indicate that FMM films enable boosting electrical properties and strain-resistant performance in fundamental logic-circuit applications.

To demonstrate the versatility of this strategy in constructing FMM structure for CP systems, we applied this strategy to a typical p-type semicrystalline conjugated polymer, DPPTT, to fabricate the hybrid film with 20% PBAT. The morphology and electrical performance of DPPTT-based hybrid film were investigated. For comparison, the neat DPPTT film and LS film comprising DPPTT and SEBS were also prepared. Similar with the morphology in N2200-based hybrid films, DPPTT-based hybrid film comprising PBAT exhibited a desired FMM structure, which is contributed to the aggregation effect during phase separation. While the LS film of DPPTT/SEBS exhibited an ordinary mesh structure with larger pore diameters, resulting from the significant phase separation (Supplementary Fig. 33). Electrical performance of stretchable OFETs employing the DPPTT-based neat film, LS film and FMM film was investigated under tensile strain from 0% to 100% (Supplementary Fig. 34). Compared to the neat film and LS film, the

FMM film exhibited higher electrical stability, especially when large strain was applied. These results demonstrated the versatility of our strategy in CP and reconfirmed the beneficial effect of FMM structure on electrical stretchability.

## Stretchable synaptic phototransistors leveraging the retina-like structure

Leveraging the retina-like structure of FMM films, stretchable synaptic phototransistors were fabricated to imitate the light-sensitive function of photoreceptor cells and synaptic signal transmission. In the visual system, photons stimulate the retina, generating action potentials as presynaptic spike, which is then transferred by neurotransmitters to postsynaptic site and triggers excitatory postsynaptic current response (EPSC)[39–44]. Information is received by the visual system through learning and forgetting process, correlated to learning time. Repeated rehearsals can consolidate the learning process and transition from short-term memory into long-term memory. This behavior can be well imitated by the synaptic phototransistor[16,45–48]. Under irradiation, light-induced excitons separate into holes and electrons, further leading to an increase in long-lasting on-current in the phototransistor, which performs a typical synaptic characteristic (Fig. 6a, b).

DPPTT was adopted to fabricate FMM films as the charge channel of optoelectrical synapse, owing to its ambient stability[49–52] and high sensitivity to visible and near-infrared light (Supplementary Fig. 35). Visible light at 760 nm was used as input signals to train the phototransistors to function analogously to the human brain. To assess the neuromorphic performance of FMM film-based phototransistors, key synaptic behaviors including EPSC, paired-pulse facilitation (PPF), short-term plasticity (STP), and long-term potentiation were investigated[20,53]. As shown in Fig. 6b, when triggered by a paired pulse with a pulse width of 5 s, an accumulating EPSC was observed, as a result of residual charge carriers after the first irradiation. The PPFs were investigated under varying intervals to validate the double-exponential function behavior (Fig. 6c, Supplementary Fig. 36). The fitting curve indicates a rapid decay time ($t_1$) of 3.85 s and a slow decay time ($t_2$) of 631.4 s. When irradiated at a short interval of 0.5 s, the memory devices demonstrate a high PPF index of 1.6, which declines to 1.2 at a long interval of 60 s. This result demonstrates that the FMM film-based phototransistors can well imitate descending learning efficiency as the intervals between successive learning stimuli increase. Besides, with increasing pulse width from 1 s to 20 s, a significantly higher residual current of 7.5 nA was observed, signifying a transformation from volatile to nonvolatile memory via signal width modulation, which well imitates the adjustable synaptic plasticity of human brain from short-term memory to long-term memory[43]. Notably, this nonvolatile memory gradually decays to the original state after 200 s (Supplementary Fig. 37). Meanwhile, synaptic phototransistors employing FMM films exhibit a twofold photoresponsivity compared to the neat film-based phototransistors (Fig. 6e, Supplementary Fig. 38). In contrast, LS film exhibited reduced photo sensitivity compared to neat film. These results indicate that FMM structures enhance the light harvesting process via the wide-angle light absorption of nanofibers. To explore the application of FMM synaptic transistor in low-energy consumption devices, a stretchable synaptic transistor with channel length of 20 μm was constructed. A consumption of 4.5 nJ was obtained by applying optical pulse of 0.5 mW $cm^{-2}$ at $V_{DS} = -1$ V (Supplementary Fig. 39), which is rather low for synaptic transistors employing only CP as photoactive material (Supplementary Table 7).

To evaluate the reliability of the synaptic phototransistor against stretching, PPF and EPSC responses to 50 successive light pulses under various tensile strains were investigated. Synaptic transistor of FMM film achieved a very stable PPF index, where only a deviation of 6.7% was observed after stretching to 100% strain (Fig. 6e). This is rather low compared to the neat-film-based synaptic transistors and

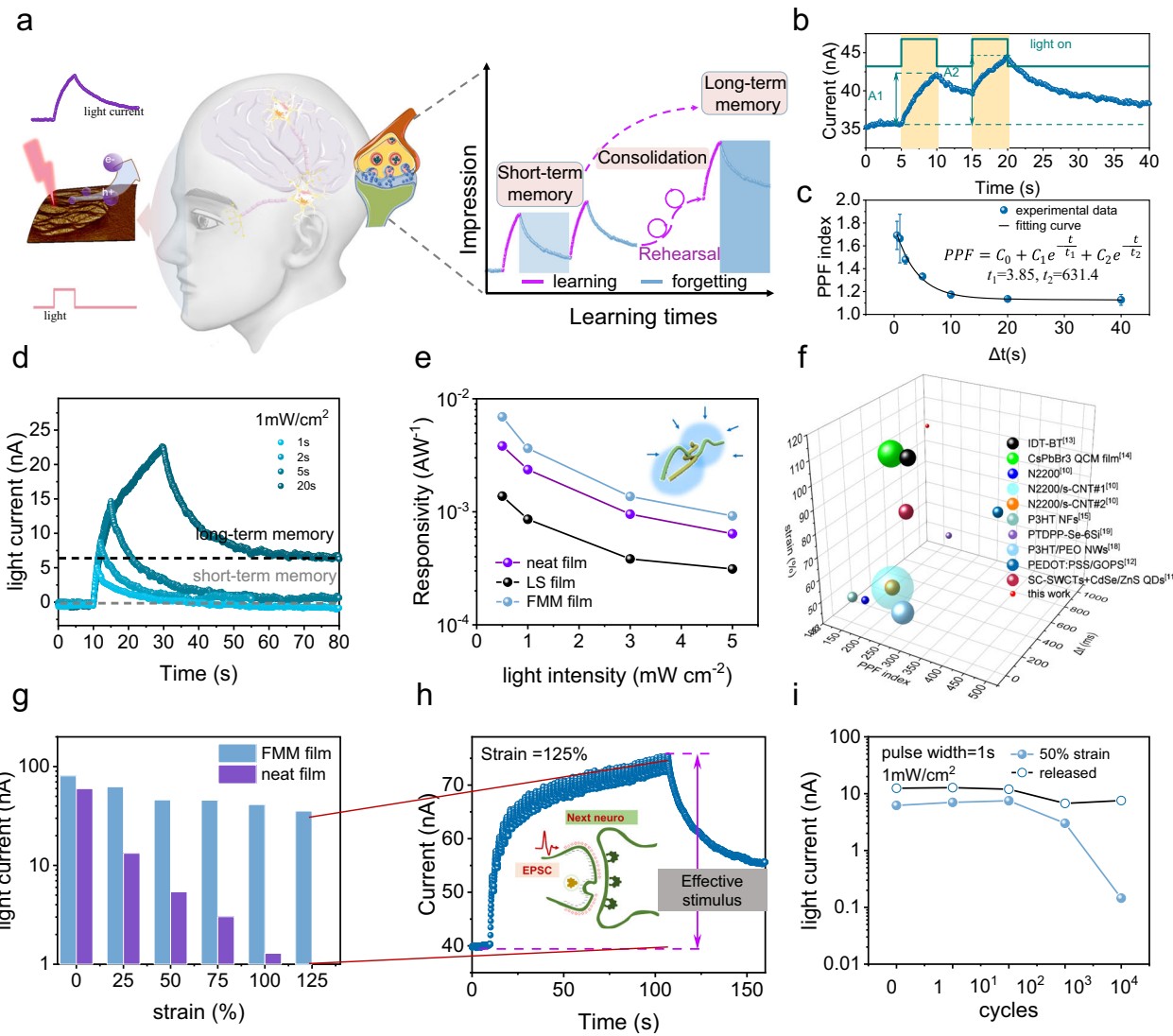

**Fig. 6 | Stretchable synaptic phototransistors employing FMM films of DPPTT imitate the visual memory in human brains. a** Schematic of the working principle in synaptic phototransistors for imitating visual memory, including the learning and forgetting process with adjustable plasticity from the short-term memory and long-term memory. **b** A typical light current response curve in the stretchable synaptic phototransistors employing FMM film of DPPTT. The $V_G = 0$ V, $V_{DS} = -60$ V. **c** PPF index of FMM synapse plotted as a function of spike intervals. Data are presented as mean values ± SD, $n = 3$. Data were obtained by three individual tests. **d** EPSC of FMM synapse under different irradiation durations. **e** Comparison of the photoresponsivity of FMM-film, LS-film and neat-film synapses under various light intensities. Calculation details are provided in Supplementary Fig. 38. **f** Comparison of stretchability, PPF index and stability of PPF index achieved by reported stretchable organic synaptic transistors and our work. IDT-BT[13], CsPbBr3 QCM film[14], N2200[10], N2200/s-CNT[10], N2200/s-CNT[10], P3HT NFs[15], PTDPP-Se-6Si[19], P3HT/PEO NWs[18], PEDOT:PSS/GOPS[12], SC-SWCTs+CdSe/ZnS QDs[11]. For each data point, the sphere radius was scaled by the deviation of PPF index ($r \propto \frac{\triangle PPF}{PPF^{0\%}}$, $\triangle PPF$ is the difference between PPF index under maximum strain and that of initial state). **g** EPSC of FMM- and neat-film synapse after 50 pulses under various strains along the charge transport, using pulse width of 1 s and light intensity of 1 mW cm⁻². **h** A representative EPSC of FMM-based synapse under strain of 125% along the charge transport. Insert: schematic of the action current transferred along the neurons, which demands critical level of the stimulus. **i** ESPC under cycling stretched and released state. Stretching direction is along the channel.

previously reported stretchable organic synaptic transistors. (Supplementary Fig. 40, Supplementary Table 8). Under increasing strain from 0% to 100%, the FMM synaptic transistors preserved 29% of the paired-pulse facilitated current, significantly surpassing the performance of LS and neat film devices. Meanwhile, an EPSC as high as 35 nA was obtained when stretched up to 125% strain (Fig. 6g, Supplementary Fig. 41). In contrast, the photocurrent of neat film-based phototransistors dramatically reduced by 90% when stretched to 100% strain, highlighting the advantages of FMM structures in stretchable device applications (Supplementary Fig. 42). To demonstrate the cycle stability of devices, synaptic phototransistors were characterized under 50% strain for 10,000 stretching-releasing cycles. FMM synaptic transistors exhibited stable device

functionality at stretching state up to 1000 cycles and reversible performance at released state after 10,000 cycles (Fig. 6i, Supplementary Fig. 43). However, the LS synaptic transistor failed to exhibit light response after 1000 cycles. Through incorporating stretchable organic semiconducting films featured with FMM structure, the synaptic phototransistors demonstrate significant photo responsivity, stretchability, and a remarkable capacity to imitate multiple biological synaptic behaviors, exhibiting great potential for developing next-generation artificial intelligence visual systems.

In summary, we have proposed an effective method of processing ultra-stretchable and high-performance OSC through VDWs-force-tuning phase separation, which is achieved by blending CP with a strongly aggregated plastic polymer PBAT. This strategy

replicates a highly functionally aligned biomimetic substructure that has rarely been reported in CPs: FMM with record-low substructure dimension and interpenetrating network. The geometry-deformable and interpenetrating micromesh improves overall stretchability of FMM film. Meanwhile, the fibril substructure enables semiconducting layers with two-fold improved photosensitivity in optoelectronics. Importantly, our results point out the importance of intermolecular interactions in tailoring the morphology of hybrid CP films and developed strategy of processing biomimetic stretchable organic semiconductors. Furthermore, combing experimental validation and FEA simulation, we revealed the dependence of strain dissipation mechanism on pore diameter, and identified a critical threshold of pore diameter to guide effective processing. Stretchable inverters and photo synapses employing FMM films have been demonstrated to exhibit high stretchability and stable performance in signal conversion and imitation of visual memory system. Thus, we expect great potential of this strategy in fabricating high-performance stretchable OSC for advanced bioelectronics.

## Methods

### Materials
Poly ([N, N-bis (2-octyl dodecyl)-naphthalene-1,4,5,8-bis(dicarboximide)-2,6-diyl]- alt-5, 5'-(2,2'-bithiophene)) (N2200) was purchased from Solarmer Ltd., Thieno[3,2-b] thiophene-diketopyrrolopyrrole (DPPTT) was purchased from Derthon. The number-averaged molecular weight ($M_n$), weight-averaged molecular weight ($M_w$), and polydispersity index (PDI) of N2200 and DPPTT were 60.3 kDa, 100.5 kDa and 1.7, and 30.3 kDa, 52.5 kDa, 1.7, respectively.

Poly (butyleneadipate-co-terephthalate) (PBAT) was purchased from Sigma Aldrich with a purity >95% and used as received. The $M_n$, $M_w$, and PDI of PBAT were 5.48, 8.18 and 1.49, respectively.

SEBS (H1052, Asahi Kasei), with a volume fraction of poly(ethylene-co-butylene) = 80%, was chosen as the dielectric layer and insulator introduced to the OSC films. The responding $M_n$, $M_w$, and PDI were 26.8, 28.7, and 1.07, respectively.

Poly(dimethylsiloxane) (PDMS, Sylgard 184) was purchased from Dow Corning.

Au was purchased from Sinopharm, with purity higher than 99.999%.

Ag nanowires were purchased from XFNANO company with a concentration of 5 mg/ml.

Octadecyltrichlorosilane (OTS) was purchased from Sinopharm with purity >85%.

Ethanol, n-hexane, Toluene and Chloroform were purchased from Sinopharm with purity of 99.7%.

$H_2SO_4$ solution was purchased from Sinopharm with weight percentage of 98%.

$H_2O_2$ was purchased from Sinopharm with weight percentage of 30%.

All the solvents were utilized as received.

### Thin films preparation
**FMM film.** N2200 and PBAT were dissolved in chloroform and stirred for 6 h (temperature: 45 °C). The resultant solution was spin-coated (1500 rpm for 45 s) on the Si/SiO$_2$ substrate and then annealed at 120 °C for 20 min.

**LS film.** Films were prepared followed by the same strategy as FMM film except for the insulator (SEBS). In section "Introduction", hybrid films with various weight ratios were studied to demonstrate the impact on pattern evolution.

**The dielectric layer.** SEBS was dissolved in methylbenzene with a concentration of 60 mg ml$^{-1}$. The films were fabricated by spin-coating

the SEBS (H1052) solution on the Si/SiO$_2$ with a speed of 1500 rad per minute for 45 s.

**Stretchable electrodes.** Both the gate and source/drain electrodes were fabricated by spray-coating the Ag nanowire solution (AgNW, dispersed in isopropyl alcohol with a concentration of 5 mg ml$^{-1}$) on the silicon wafer through a shadow mask. All the doped silicon wafers with a SiO$_2$ dielectric (300 nm) are treated to form a highly hydrophobic surface. In detail, octadecyltrichlorosilane (OTS) was dissolved in n-hexane with concentration (v/v) of 1:100. Si/SiO$_2$ plates are soaked in the solution for 2 min to form a layer of OTS with the hydrophobic surface.

PDMS substrate was prepared at a ratio of 20:1 (base/cross-linker, w/w) and cured at 80° C on a slide for 2 h.

### Device construction
The rigid field effect transistors were fabricated on the Si/SiO$_2$ with a bottom-gate-top-contact structure. First, the Si/SiO$_2$ substrates were treated with O$_2$ plasma for 2 min and then modified with OTS solution (diluted using n-hexane, OTS/n-hexane (v/v) =1:200) for 60 s. Then, the semiconducting polymer solutions were cast on the Si/SiO$_2$ substrates and annealed in the N$_2$ glove box for 30 min. Finally, the Au was fabricated on the semiconducting polymer layers through thermal evaporation. The thickness of Au electrodes was 40 nm. The channel width was 1500 μm, the channel width was 30 μm ad 50 μm, respectively.

The fully stretchable OFETs were fabricated by utilizing the PDMS substrate to pick up the upper layers with the order of gate electrode, dielectric layer, semiconducting polymer films, and source and drain electrodes.

### Characterization
AFM measurement was conducted on the Bruker atomic force microscopy. Optical microscope images were collected on the Nikon optical microscope. UV–vis absorption spectrum was collected from the SHIMADZU (2600). GIXRD was conducted at Shanghai Synchrotron Radiation Facility (SSRF) at beamline 14B/15U. Numerical integration of the diffraction peak areas was performed using Dioptas and Fit2d. The contact angle measurement was performed on the contact angle meter (Dataphysics OCA20).

**Optoelectrical characterization.** The electrical characterization of n-type OFETs were carried out by the PDA FS380 in the N$_2$ glove box. The performance of the synaptic transistors was measured by the Keithley 4200 in ambient conditions utilizing the 760 nm laser as a light source (Changchun New Industry Photoelectric Technology Co., Ltd.).

The capacitance of the stretchable OFETs was measured in a plate capacitor employing AgNW stretchable electrodes and SEBS (H1052) dielectric layer (thickness ~2 μm). The overlapped area of the bottom and upper electrodes was 0.01 cm$^2$ at initial state, and changes with the strain, which is consistent with the stretchable OFETs. Capacitance was an average value of at least three devices measured under 1000 Hz by PDA FS380 in the N$_2$ glove box.

## Data availability
The data that support the findings of this study are available in the Supplementary Information. Source data are provided with this paper. All data are available from the corresponding author upon request. Source data are provided with this paper.

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

## Acknowledgements

This research was financially supported by the National Key R&D Program of China (No. 2022YFB3603804), the National Natural Science Foundation of China (Grant Nos. 52473297) received from Y.Z., and the Natural Science Foundation of Shanghai (24ZR1480800) received from Z.W. The authors gratefully acknowledge the Shanghai Synchrotron Radiation Facility (SSRF) for providing the precious time and the Synchrotron XRD Facility at Beamline NO. 14B/15U.

## Author contributions

Q.Z. and Y.Z. conceived the idea and designed the experiment. Y.L. supervised the project. Q.Z. performed all the experiments and analysed all the experimental data. Q.Z. and Z.W. wrote the manuscript. Y.Z. reviewed the paper. X.X., G.Z., W.L., H.Z., L.S. gave valuable comments on this paper.

## Competing interests

The authors declare no competing interests.
