## [Transparent Peer Review file · Nature Communications]

Biomimetic Fibrous Semiconducting Micromesh via Tuning Phase Separation for High-Performance Stretchable Optoelectronic Synapses

Corresponding Author: Professor Yan Zhao

Version 0:

Reviewer comments:

Reviewer #1

(Remarks to the Author)

The peer-reviewed paper entitled "Biomimetic Fibrous Semiconducting Micromesh via Tuning Phase Separation for High-Performance Stretchable Optoelectronic Synapses" describes a methodology for obtaining strong and stretchable materials based on conducting polymers. The manuscript describes in detail the synthesis and strength studies of the materials, as well as their application as optoelectronic synapses. In my opinion, the presented work describes interesting studies on fibrous semiconducting micromesh materials, describing in detail the characteristics of these materials. I believe that the presented approach and obtained results are interesting, but this work requires refinement of several aspects.

In my opinion, the presented manuscript presents a very interesting topic, but it is written in an inconsistent way and is difficult to read. It is not entirely clear whether this work is supposed to concern a new method of obtaining polymer materials, ultra-stretchable materials or synapses, e-skin or visual memory system.

Below I present detailed comments.

1. The theoretical introduction requires improvement and clarification of what this article is about. The introduction begins with a description of e-skin - its applications and properties of polymers used for its synthesis, and in the rest of the article the authors focus rather on the application of the studied materials to build an optoelectronic synapse. In Chapter 2.6 this material is described as inspired by a retinal-like structure. Also TOC suggests that the article will deal with this retina-like structures. The authors should specify more precisely in the introduction what applications their material can be used for. In addition, the introduction uses several general statements, e.g. "The films exhibit high electrical stretchability and two-fold enhancement in photosensitivity" (line 81-81) and it is not known for which material a two-fold increase was observed.
2. The part describing the concept of manufacturing the polymeric material (chapter 2.1, lines 100-107) is not very clear. Please explain which polymer plays a specific role in the system and clarify the descriptions of the corresponding elements in Figure 1a. It is not fully understood why a reference to Figure S1, which compares the results obtained for PBAT and SEBS, is given here, because the text does not mention SEBS. Please improve this part.
3. There are a lot of abbreviations throughout the manuscript, which makes it much harder for the reader to follow the content. I would suggest making a list of abbreviations, which will make it easier to find your way around. Some abbreviations are not explained, e.g. P(NDI2OD-T2) (N2200) (line103). It would be a good idea to add there a reference to Figure 1a, where the structure of this polymer is drawn.
4. In the manuscript there is often a comparison of results obtained for 3 types of thin films. It would be appreciated for easier understanding of the topic if there was some diagram illustrating the difference in the structure of these films.
5. The presented results of neuromorphic measurements are not complete. First, Figure 6e shows a comparison of the dependence of the generated photocurrent for different intensities of incident light for two films: neat and FMM. The supporting information in Figure S36 shows the results only for the neat film, but there is no data for the FMM film. Please attach the results of analogous measurements for FMM film.
6. When presenting the PPF repeatability results of the obtained signals for different strain tensors in Figure S37, the scales and values for the neat and FMM films are presented at different scales and measured for different conditions. Furthermore, it is not described how the results presented in the plots a-d differ.
7. The manuscript also lacks information on how long the long-term memory state lasts and whether the system returns to its original state after a longer period of time or not. And if not, is it related to material degradation? In Figure 6d, only 80 s is shown as the long-term memory time. What is the repeatability of the results obtained in PPF measurements?

I also have some comments regarding the quality of the presented figures and tables:

1. The TOC figure is very simple and not very attractive to the reader of Nature Communication. I believe that it does not fully reflect the value of the article.
2. In Figure 1f and 3 b and 3 c the pore diameters are shown. In both figures different units are shown. In Figure 1f the pore diameter is expressed in μm^{-1} , while in Figure 3b and 3c this value is expressed in nm. It would be good to unify these units.
3. In Figure 3, the descriptions on the AFM photos are in white and are difficult to see. I suggest changing the descriptions to black, for example.
4. Below Table 1, line 446 describes the μsat value, which is not in the table, but it does not explain what the other symbols mean: μsat , V_{th} . Please fill this in.

The text also contains minor linguistic errors, including:

- Line 59 "N-type semiconductors" n is usually written in lower case, like in line 403
- Line 416 "P-type semicrystalline" p is usually written in lower case, like in line 403
- Line 79 Van de Waals force
- Line 106 Van der Waal's,
- Line 380 should be Figure 5a instead 4a
- Line 453 it should be EPSC instead ESPC

I believe that after taking into account all the corrections the article can be published.

Reviewer #2

(Remarks to the Author)

The authors have fabricated biomimetic hybrid semiconducting films featuring geometry-deformable micromesh and nanofibril substructure via tuning phase separation. The films exhibit enhanced stretchability. The authors have fabricated synaptic phototransistors with superior synaptic plasticity and robust performance under strains up to 125% and 1000 repeated cycles at 50% strain. However, the following issues should be addressed.

1. Could the authors discuss the underlying mechanism of Van der Waals force driving PBAT chains to form ribbon-like L-L separation at nanoscale?
2. Figure 6b shows the synaptic plasticity of the stretchable synaptic phototransistors. The VDS is high (-60V), which may lead to high energy consumption. Do the authors have a solution to this problem?
3. The authors claim that the synaptic phototransistors demonstrate significant photosensitivity. The authors need to measure typical parameters related to photosensitivity.
4. The authors should compare the current device with previously reported stretchable synaptic devices in terms of energy consumption, stretchability, etc.?

Reviewer #3

(Remarks to the Author)

In their manuscript, Zhou et al. introduced a stretchable semiconductor film featuring a geometry-deformable micromesh and nanofibril substructure, achieved through phase separation mediated by Van der Waals forces. This design, which combines the geometry-deformable micromesh and an interpenetrating phase within the blended semiconductor-poly(butylene adipate-co-terephthalate) (PBAT) film, enhances both the mechanical and electrical stretchability of the semiconductor. As a result, this fibrous stretchable semiconductor shows potential for applications in stretchable optoelectronic synaptic devices. This work is interesting, but there are several issues need to be addressed.

- (1) This work need more demonstration to show, how their approach significantly advances the field. Similar methodologies have been explored in existing literature, and the manuscript does not highlight distinct innovations or improvements.
- (2) Moreover, the experimental design lacks critical controls, making it difficult to validate the claims of improved performance. The demonstration of the device's synaptic performance is also weak, as it fails to maintain stable performance under deformation and exhibits only basic synaptic functionality.
- (3) The design of micromesh or nanoconfinement structures through precise control of phase separation in blend films of semiconductors and elastomers is well documented in previous works [e.g., Science, 2017, 355, 59; Nat. Electron. 2022, 5, 881]. The phase separation ratio can be effectively tuned by adjusting the ratio of the polymer semiconductor to elastomers. These methods have also been successfully employed in the development of stretchable optoelectronics [e.g., Chem. Eng. J., 2024, 492, 152143]. The authors claim they can precisely control the Van der Waals forces, but their method of tuning phase separation by changing the ratio of the semiconductor to PBAT to control fiber and pore diameters does not appear to differ from existing works [e.g., Science, 2017, 355, 59].
- (4) The comparison between the FMM film (i.e., N2200-PBAT film) and the LS film (N2200-SEBS film) is unfair. The performance of the FMM film is optimized for the best PBAT/N2200 ratio, whereas the LS film, although using the same weight ratio as the FMM film, may not be at its optimal SEBS/N2200 ratio. The authors should also determine the optimal ratio for the LS film and then compare the performance improvements of their FMM film-based device to the best LS film-based device. Additionally, the authors should fabricate more devices using various polymer semiconductors and compare their performance. This will help verify that their method is broadly applicable and represents an advancement over existing techniques. Comparing only N2200-based FMM film with N2200-based LS film is not sufficiently convincing.

- (5) The demonstrated synaptic phototransistor performance is not very promising, as similar results have already been widely reported in the literature. Additionally, the strain significantly affects the device's synaptic performance, making it challenging to use in practical applications. Moreover, the comparison of synaptic performance is only between the neat film and the FMM film. Why not include the synaptic performance of the LS film? Is the synaptic performance of the FMM film superior to that of the LS film?
- (6) The change in the PFF index with pulse time interval is unusual. As shown in Figure 6c, the value initially shows a very fast decrease, and then slightly increases as the time interval increases. Why is that?
- (7) In Figure 6d, the final device current state after 1 second of irradiation is larger than that after 2 seconds and 5 seconds of light irradiation. Could you explain the reason for this? It appears that the device state after light irradiation does not fully return to the initial state. Can the memory states of this device be erased?
- (8) How stable is the device under long-term light irradiation? How many cycles can the device operate during repeated light on and off conditions? Additionally, how consistent is the EPSC value across different test cycles for the same device, and how repeatable is it from one device to another?
- (9) The crack onset strain (COS) test depicted in Figure 1e should include error bars to enhance the accuracy of the results. Additionally, it is important to specify the substrate used, as it can significantly impact the COS value.
- (10) The stretching cyclic stability of the FMM-based device should be compared to that of the LM-based device.
- (11) The film's stability has not been adequately addressed. It is essential to evaluate whether the film morphology will change over prolonged storage durations. According to a study published in Science [Science, 2017, 355, 59], nanoconfinement films have demonstrated the capability to sustain stable device performance when stored for up to one year. Therefore, the authors are urged to explore the stability of both their film and devices over extended storage periods and under varying humidity and temperature conditions.
- (12) The use of the term "biomimetic" in the title may be misleading. While I acknowledge that the film exhibits an abundance of fiber-like structures, it is challenging to draw a direct connection between these structures and the complex photoreceptor cells found in the eyes, as there is no functional similarity. Additionally, the photoresponse primarily originates from the semiconductor itself rather than from the fiber-like structure design.

Version 1:

Reviewer comments:

Reviewer #1

(Remarks to the Author)

All my comments and remarks have been responded to. I accept all corrections and recommend the manuscript for publication.

Reviewer #2

(Remarks to the Author)

The authors have largely addressed all the comments. I would like to recommend the publication of this revised manuscript.

Reviewer #3

(Remarks to the Author)

The revision addressed all questions.

RESPONSES TO THE COMMENTS OF THE REVIEWERS

Comments in *blue* – Replies in black

The changes have been highlighted in the revised manuscript with their page or figure numbers mentioned in the replies.

RESPONSES TO THE COMMENTS OF THE REVIEWER #1

Comment: The peer-reviewed paper entitled "Biomimetic Fibrous Semiconducting Micromesh via Tuning Phase Separation for High-Performance Stretchable Optoelectronic Synapses" describes a methodology for obtaining strong and stretchable materials based on conducting polymers. The manuscript describes in detail the synthesis and strength studies of the materials, as well as their application as optoelectronic synapses. In my opinion, the presented work describes interesting studies on fibrous semiconducting micromesh materials, describing in detail the characteristics of these materials. I believe that the presented approach and obtained results are interesting, but this work requires refinement of several aspects.

In my opinion, the presented manuscript presents a very interesting topic, but it is written in an inconsistent way and is difficult to read. It is not entirely clear whether this work is supposed to concern a new method of obtaining polymer materials, ultra-stretchable materials or synapses, e-skin or visual memory system.

Below I present detailed comments.

Response: We are grateful to the reviewer's comments and suggestions and appreciate the positive recommendation for publication as a novel and interesting work. We have revised the manuscript carefully according to the comments, as listed below.

1. The theoretical introduction requires improvement and clarification of what this article is about. The introduction begins with a description of e-skin - its applications and properties of polymers used for its synthesis, and in the rest of the article the authors focus rather on the application of the studied materials to build an optoelectronic

synapse. In Chapter 2.6 this material is described as inspired by a retinal-like structure. Also TOC suggests that the article will deal with this retina-like structures. The authors should specify more precisely in the introduction what applications their material can be used for. In addition, the introduction uses several general statements, e.g. "The films exhibit high electrical stretchability and two-fold enhancement in photosensitivity" (line 81-81) and it is not known for which material a two-fold increase was observed.

Response: Thank you very much for your valuable comments.

In this work, we are providing a new method of obtaining ultra-stretchable semiconducting materials, which can be applied in optoelectronic devices, such as OFETs and synaptic transistors. In the introduction part of the revised manuscript, we specified that we are developing processing strategy for polymer semiconductors integrated to photonic synaptic transistors. The details revised on Page 2 of the manuscript are listed below.

“Among them, OFET-based optoelectronic synapses are gaining attentions for the potential to emulate visual perception, adaptation, and imaging. In particular, the development of photo-responsive synapses combining high stretchability will booth advancements in many fields. Despite its promises, most of the reported synaptic OFETs exhibited degradable performance under deformation, due to the lack of conjugated polymers with high optoelectronic performance and stretchability.”

And in line 76-78, we specified that

“The FMM structure imparts high electrical stretchability to the polymer semiconductors and leads to a two-fold enhancement in photoresponsivity in the DPP-TT-based FMM film.”

2. The part describing the concept of manufacturing the polymeric material (chapter 2.1, lines 100-107) is not very clear. Please explain which polymer plays a specific role in the system and clarify the descriptions of the corresponding elements in Figure 1a. It is not fully understood why a reference to Fig.1, which compares the results obtained for PBAT and SEBS, is given here, because the text does not mention SEBS. Please improve this part.

Response: We appreciate the reviewer's insightful comment regarding section 2.1.

We have carefully revised this section to clearly indicate the specific roles of each polymer component in the system. In section 2.1, we now explicitly state that PBAT serves as stretchable component, while N2200 acts as semiconducting component. In addition, we have revised the labels of materials in Figure 1a, by adding "Stretchable component" and "Semiconductor" before the name of PBAT and N2200, respectively. The revised details are listed as below.

"After fabrication, the interpenetrating network of OSC and PBAT is formed in the hybrid film, where the continuous OSC provides charge transport path and PBAT serve as the stretchable component under deformation."

Regarding the reference to Fig.1, we acknowledge the reviewer's concern. We have removed this reference here and refer Supplementary Fig.1 in line 107-109, while introducing the stretchability of PBAT.

"Differing from commonly used elastic insulators which exhibit amorphous structure, PBAT possesses high stretchability based on a semicrystalline structure (Supplementary, Fig. 1)"

We hope these revisions improve the clarity of the manuscript and make the manufacturing process more accessible to readers.

3. There are a lot of abbreviations throughout the manuscript, which makes it much harder for the reader to follow the content. I would suggest making a list of abbreviations, which will make it easier to find your way around. Some abbreviations are not explained, e.g. P(NDI2OD-T2) (N2200) (line103). It would be a good idea to add there a reference to Figure 1a, where the structure of this polymer is drawn.

Response: We sincerely thank the reviewer for the helpful suggestion.

We have carefully reviewed the manuscript to ensure that all abbreviations, including N2200, are clearly defined at their first appearance. Additionally, we have now added a list of abbreviations at the end of the revised manuscript (Page 25) to improve readability and make it easier for readers to navigate the text. Furthermore, we have added a reference to Figure 1a in the revised manuscript (Page 4), where the

chemical structure of N2200 is shown, as suggested. As listed below:

“Chemical structures of PBAT and N2200 are listed in Figure 1a.”

We believe these changes enhance the clarity and accessibility of the manuscript for readers.

4. In the manuscript there is often a comparison of results obtained for 3 types of thin films. It would be appreciated for easier understanding of the topic if there was some diagram illustrating the difference in the structure of these films.

Response: We sincerely thank the reviewer for this helpful suggestion.

To improve clarity and assist readers in understanding the structural differences among the three types of semiconducting films, we have added a schematic diagram in the revised manuscript (Supplementary Fig. 21). This figure illustrates the key distinctions in film morphology and phase composition among the films and is properly referenced in the relevant section in the manuscript line 337-338. The diagram is as listed below.

“For clarity, the structural difference of neat film, LS film and FMM films are summarized in Supplementary Fig. 21.”

Structural difference		Films		
		neat film	FMM film	LS film
Composition		N2200	N2200/PBAT (6:4)	N2200/SEBS (6:4)
Film morphology		Dense	Mesh structure	Mesh structure
		Non-fibrous	Fibrous	Non-fibrous
		SEM		adhesion				
Phase composition		N2200 	Interpenetrating phase 	Macro phase separation 
Supplementary Figure 21. Structural differences of neat film of N2200, FMM film and LS film. For the FMM film and LS film, the conjugated polymer is N2200, and the insulator component is PBAT and SEBS, respectively.

5. The presented results of neuromorphic measurements are not complete. First, Figure 6e shows a comparison of the dependence of the generated photocurrent for different intensities of incident light for two films: neat and FMM. The Supplementary information in Fig.36 shows the results only for the neat film, but there is no data for the FMM film. Please attach the results of analogous measurements for FMM film.

Response: We sincerely apologize for the oversight and appreciate the reviewer for pointing this out.

In the revised manuscript, we added the analogous measurements for FMM film and LS film. Correspondingly, we updated the photoresponsivity of the neat film, LS film and FMM film of DPP-TT in Figure 6e. As listed below.

Supplementary Figure 40. Photoelectric responses of the synaptic transistors employing (a) FMM and (b) neat film and (c) LS film of DPP-TT under varying irradiation intensities.

Figure 6e. Comparison of the photoresponsivity of FMM-film, LS-film and neat-film synapses under various light intensities.

6. When presenting the PPF repeatability results of the obtained signals for different strain tensors in Fig.37, the scales and values for the neat and FMM films are presented at different scales and measured for different conditions. Furthermore, it is not described how the results presented in the plots a-d differ.

Response: Thank you very much for your comments.

In the revised manuscript, we updated the data in Supplementary Fig. 37 (Supplementary Fig. 42 in the revised manuscript), which were measured under the same condition. For synaptic transistors based on different kinds of films, the dark current and EPSC are at different levels. Thus, into order to make the data more accessible, we presented the data at different scales. In the revised manuscript, we added EPSC of LS-synaptic transistors under uniaxial stretching, as a comparison of FMM-

synaptic transistors. We also extracted the PPF index of synapses under strain in Supplementary Fig. 42. Meanwhile, in the revised manuscript (line 515-521), we discussed the difference of device performance. As listed below.

“Synaptic transistor of FMM film achieved a very stable PPF index, where only deviation of 6.7% was observed after stretching to 100% strain (Figure 6e). This is rather low compared to the neat-film based synaptic transistors and previously reported stretchable organic synaptic transistors. (Supplementary Fig. 42, Supplementary Table 5). Under increasing strain from 0% to 100%, the FMM synaptic transistors preserved 29% of the paired-pulse facilitated current, significantly surpassing the performance of LS and neat film devices.”

Supplementary Figure 42. EPSC under paired pulses of the stretchable synaptic phototransistors under strain. (a) Synaptic transistor employing neat film, (b) Synaptic transistor employing FMM film; (c) Synaptic transistor employing LS film. (d) PPF index of the stretchable synaptic transistors under strain. The irradiation power was 10 mW cm^{-2} . The wavelength was 760 nm .

Figure 6f. Comparison of stretchability, PPF index and stability of PPF index achieved by reported stretchable organic synaptic transistors and our work. For each data point, the sphere radius was scaled by the deviation of PPF index ($r \propto \frac{\Delta PPF}{PPF^{0\%}}$, ΔPPF is the difference between PPF index under maximum strain and that of initial state).

Supplementary Table 5. Comparison of the PPF index of reported stretchable synaptic phototransistors before and after stretching.

reference	Semiconducting layer	PPF index (%)	Δt (ms)	PPF index (%) at stretched states	$\frac{\Delta PPF}{PPF^{0\%}}$ (%)
[14]	IDT-BT	191	500	123 ($\epsilon=100\%$)	35.6
[8]	CsPbBr ₃ QCM film	148	500	232 ($\epsilon=100\%$)	56.7
[9]	N2200	177	50	157* ($\epsilon=50\%$)	-11.2
[9]	N2200/s-CNT	200	250	2.14* ($\epsilon=50\%$)	114
[9]	N2200/s-CNT	199	250	1.32* ($\epsilon=50\%$)	-33.6
[10]	P3HT NFs	146	40	114* ($\epsilon=50\%$)	-22
[11]	PTDPP-Se-6Si	185	1000	167* ($\epsilon=50\%$)	-10
[13]	P3HT/PEO NWs	273	50	146* ($\epsilon=50\%$)	-47
[6]	PEDOT:PSS/GOPS	477	132	380* ($\epsilon=100\%$)	-20
[7]	SC-SWCTs+CdSe/ZnS QDs	206	400	141* ($\epsilon=80\%$)	-32
This work	DPP-TT/PBAT FMM film	130	1000	138 ($\epsilon=100\%$)	6.7

The “*” in table indicated that the PPF index were not published directly and were derived by the given equations and graphs.

7. The manuscript also lacks information on how long the long-term memory state lasts and whether the system returns to its original state after a longer period of time or not. And if not, is it related to material degradation? In Figure 6d, only 80 s is shown as the long-term memory time. What is the repeatability of the results obtained in PPF measurements?

Response: Thank you for your valuable comments.

To verify whether the system returns to its original state after irradiation of 20s, we measured the EPSC after irradiation for over 200s, and according to the results, the EPSC returned to the original state after 200 s. We added the information in the revised manuscript line 504 and in the Supplementary Fig. 39. As listed below.

“Notably, this nonvolatile memory gradually decays to the original state after 200 s (Supplementary Fig. 39).”

Supplementary Figure 39. EPSC of FMM synapse after 20 s irradiation. The current returned to the original state after 200 s.

The PPF index in Figure 6c is averaged from 3 independent measurements. We mentioned this information in the revised manuscript line 547.

Minor revision in response to REVIEWER 1

1. The TOC figure is very simple and not very attractive to the reader of Nature Communication. I believe that it does not fully reflect the value of the article.

Response: Thank you for your comments. We agree with the idea that a more attractive and insightful TOC figure would match this manuscript better. We refined the TOC figure, as shown below.

2. In Figure 1f and 3 b and 3 c the pore diameters are shown. In both figures different units are shown. In Figure 1f the pore diameter is expressed in μm , while in Figure 3b and 3c this value is expressed in nm. It would be good to unify these units.

Response: We sincerely thank the reviewer for this valuable suggestion.

In response, we have revised the figure to adopt a logarithmic scale for both pore size and fiber diameter. This approach preserves the ability to differentiate among various materials based on their characteristic dimensions, while ensuring consistency with the dimensional representation used in Figure 3, thereby enhancing interpretability. In this plot, X-values approaching zero indicate dense films, whereas Y-values near zero

suggest the absence of pronounced fibrous structures. The revised and original versions of Figure 1f are presented below for comparison.

Figure R1. Figure 1f before (a) and after revision (b).

3. In Figure 3, the descriptions on the AFM photos are in white and are difficult to see. I suggest changing the descriptions to black, for example.

Response: We sincerely thank you for your kind suggestions. We are very sorry that the notions on the AFM images are difficult to see in color white. Actually, we have tried other colors according to your suggestions. It turns out that color white works better than other colors for clarity. Thus, we keep the color of the notions in Figure 3a unchanged.

4. Below Table 1, line 446 describes the I_{on} value, which is not in the table, but it does not explain what the others symbols mean: μ_{sat} , V_{th} . Please fill this in.

Response: Thank you for your comments.

We checked the corresponding data and inserted the I_{on} value between “ I_{off} ” and “current retention”. Meanwhile, table legend was added at the bottom of the table to explain the meaning of the symbols.

Table 1. Parameters of the fully stretchable organic transistors.

Materials Contents	Device state	μ_{sat} ($\text{cm}^2\text{V}^{-1}\text{s}^{-1}$)	V_{th} (V)	I_{off} (nA)	I_{on} (μA)	Current retention (%)
Neat film	initial	0.16	0.26	18.9	13.3	55
	100% strain	0.28	10.6	9.68	7.35	
FMM film	initial	0.08	0.34	3.03	3.87	228
	100% strain	0.33	2.2	1.49	8.84	
LS film	initial	0.21	6.5	51.5	10.5	11
	100% strain	0.075	13.9	50.1	1.16	

^{a)} μ_{sat} represents the field effect mobility extracted from the saturation region; V_{th} represents the threshold voltage; I_{on} was extracted from the transfer curves at $V_G = 60$ V, $V_{DS} = 60$ V; I_{off} was extracted from the transfer curves at $V_G = 0$ V, $V_{DS} = 60$ V. Current retention is the ratio of I_{on} under 100% strain to I_{on} at initial state.

The text also contains minor linguistic errors, including:

Line 59 “N-type semiconductors” n is usually written in lower case, like in line 403

Line 416 “P-type semicrystalline” p is usually written in lower case, like in line 403

Line 79 Van de Waals force

Line 106 Van der Waal’s,

Line 380 should be Figure 5a instead 4a

Line 453 it should be EPSC instead ESPC

Response: Thank you for your careful checking. We have corrected the manuscript according to your comments.

- (1) We corrected the “N-type semiconductors” and “P-type semicrystalline” to “*n*-type semiconductors” and “*p*-type semicrystalline”, respectively.
- (2) In line 75 (page 3), 105 (page 4), 130 (page 5), the term “*Van de Waals force*” was corrected to “*Van der Waals force*”, and in line 106, *Van der Waal’s* was corrected to *Van der Waals force*.
- (3) In line 394 (page 15) *Figure 4a* was corrected to *Figure 5a*.
- (4) In line 480 (page 19) ESPC was corrected to EPSC.

RESPONSES TO THE COMMENTS OF THE REVIEWER #2

Comment: The authors have fabricated biomimetic hybrid semiconducting films featuring geometry-deformable micromesh and nanofibril substructure via tuning phase separation. The films exhibit enhanced stretchability. The authors have fabricated synaptic phototransistors with superior synaptic plasticity and robust performance under strains up to 125% and 1000 repeated cycles at 50% strain. However, the following issues should be addressed.

Response: We are grateful for the reviewer’s valuable comments and suggestions, which help to demonstrate the contribution of our strategy and clarify the underlying mechanism. We have revised the manuscript carefully according to each comment, as listed below.

1. Could the authors discuss the underlying mechanism of Van der Waals force driving PBAT chains to form ribbon-like L-L separation at nanoscale?

Response: We sincerely thank the reviewer for this valuable suggestion.

In the revised manuscript (page 7, section 2.2), we aimed at exploring the underlying mechanism of Van der Waals force during the formation of FMM structure. To begin with, we introduced the general model of phase separation of the hybrid polymer solution (line 180-191). After that, we discussed about the impact of Van der Waals force. We emphasized that the intensive intermolecular interactions in PBAT

enhanced the chains assembly during L-L separation, which modulates the competition between chain assembly and chain diffusion during phase separation. The discussion in revised manuscript is listed as below:

“The FMM film features with micromesh morphology constructed from nanoscale fibrils, which is dramatically different from typical cylinder sphere or lamellar structure. The intensive intermolecular interactions in PBAT played an important role in the formation of this fibrous mesh structure, via tuning the competition between chain assembly and chain diffusion. As the mixed solution concentrates, short-range interactions (Van der Waals force) induce PBAT chains to form clusters. Such localized condensation leads to minor L-L phase separation. Owing to the inherent rigidity, PBAT chains tend to exhibit long persisting length, which extends the molecular cluster along backbone direction, resulting in ribbon-like L-L separation at nanoscale.”

2. Figure 6b shows the synaptic plasticity of the stretchable synaptic phototransistors. The VDS is high (− 60V), which may lead to high energy consumption. Do the authors have a solution to this problem?

Response: We sincerely thank the reviewer for this valuable suggestion. In our manuscript, to obtain a detectable current, we applied a source drain bias of − 60 V. This high voltage leads to a high energy consumption in the phototransistors. In order to lower the energy consumption by using lower V_{DS} , the following strategies are recommended:

- (1) Replace the polymer semiconductors with high-mobility semiconductors, such as semiconducting crystals. However, insufficient stretchability is a long-standing problem in crystalline materials. Another solution is to blend high mobility semiconductors (PbS QDs, CsPbBr₃), with polymer semiconductors.
- (2) Increase the W/L of the device. This is a feasible strategy for the stretchable organic transistors, by increasing the channel or shortening the channel length. To exploring the potential of this strategy, we fabricated a set of phototransistors with varying channel length, ranging from 50 μm to 20 μm , aiming at achieving phototransistor with lower energy consumption. As shown

in Figure R2 and table R1, for all the transistors, when the V_{DS} was -60 V, light current increased from 7 nA to 50 nA with the channel length decreasing from 50 nm to 20 nm. To lower the energy consumption, V_{DS} was lowered to -1 V, light current of 0.9 nA was obtained. The energy consumption was lowered to 4.5 nJ. Thus, increasing the W/L of the device is an effective and universal strategy to construct low-consumption synaptic transistors.

- (3) Another way to increase the drain current is to improve the density of charge carrier in the channel. Previous researches confirmed that the light current could be improved by applying proper gate bias, in which way the traps in the channel could be filled by the charge carrier induced by the gate electric field.

Figure R2. (a) EPSC of FMM synaptic transistors with varying channel lengths. V_{DS} was -60 V, irradiation power is 0.5 mW/cm². Irradiation time was 5 s. (b-d) electrical response of FMM synaptic transistors under varying drain voltage, -60 V, -40 V and -1 V. The channel length was 20 µm, irradiation power was 0.5 mW/cm², the irradiation time was 5 s.

Table R1. light current and energy consumption of synaptic transistors with varying V_{DS} and channel length. Irradiation period is 5 seconds. Irradiation power is 0.5 mW cm⁻².

Channel length (nm)	50	30		20	
V_{DS} (V)	-60	-60	-60	-40	-1
I_{ph} (nA)	7	9	50	32	0.9
Energy consumption (nJ)	2100	2700	15000	6400	4.5

3. The authors claim that the synaptic phototransistors demonstrate significant photosensitivity. The authors need to measure typical parameters related to photosensitivity.

Response: Thank you for your insightful comments.

In the first version of our manuscript, we compared the photo-responsibility semiconducting polymer films via the photocurrent (I_{ph}) under the same irradiation in Figure 6e. Here, we measured the responsibility of the transistors. Responsibility (R) is defined as

$$R=I_{ph}/P$$

Where I_{ph} is the difference of drain currents between dark and illuminated states, and P is the incident power. The drain current was collected under constant irradiation condition for 5 s while the V_{DS} set to -60 V. V_{GS} was set to 0 V during measurement. The raw data is shown in supplementary figure 40.

Calculation details:

I_{dark} was collected under dark environment.

I_{light} was collected under irradiation condition.

I_{ph} is the difference between the I_{light} and I_{dark} .

The incident power was scaled by the light intensity and the irradiation area of the device. The light intensity is controlled by the light source, and the irradiation area is defined by the channel length and channel width, which is $5000 \mu\text{m} * 100 \mu\text{m}=5*10^{-3} \text{ cm}^2$.

The photoresponsivity of neat film, FMM film and LS film are summarized in **Figure 6e**.

Figure 6e. Comparison of the photoresponsivity of FMM-film, LS-film and neat-film synapses under various light intensities.

4. The authors should compare the current device with previously reported stretchable synaptic devices in terms of energy consumption, stretchability, etc.?

Response: We sincerely thank you for the thoughtful and constructive comments.

We agree that this is a good way to show the stability of our devices. We compared the energy consumption, stretchability and deviation of PPF at stretched state of our device and the reported stretchable/flexible organic synaptic transistors. The detailed data is provided in the Supplementary information table 4 and 5, and also attached as below. As depicted in the tables, our devices demonstrated leading stability. When stretched to 100% strain, the deviation of PPF in merely 7%, which is much lower than that of other reported stretchable organic synaptic transistors. However, because of the rather lower mobility of the polymer semiconductor DPP-TT, our device exhibited higher energy consumption than other stretchable organic synaptic transistors employing hybrid semiconducting materials. But the energy consumption of 4.5 nJ is rather low among stretchable organic synaptic transistors employing only conjugated polymers as photo active material.

In the revised manuscript, we discussed the performance of our FMM synaptic transistor with reported stretchable organic synaptic transistors as below:

“To explore the application of FMM synaptic transistor in low-energy consumption devices, a stretchable synaptic transistor with channel length of 20 μm was constructed. A consumption of 4.5 nJ was obtained by applying optical pulse of 0.5 mW/cm^2 at $V_{\text{DS}} = -1$ V (Supplementary Fig. 41), which is rather low for synaptic transistors employing

only conjugated polymers as photo active material (Supplementary Table 4).”

“To evaluate the reliability of the synaptic phototransistor against stretching, PPF and EPSC responses to 50 successive light pulses under various tensile strains were investigated. Synaptic transistor of FMM film achieved a very stable PPF index, where only deviation of 6.7% was observed after stretching to 100% strain (Figure 6e). This is rather low compared to the neat-film based synaptic transistors and previously reported stretchable organic synaptic transistors. (Supplementary Fig. 42, supplementary Table 5).”

Supplementary Table 4. List of previously reported organic synaptic phototransistors with our FMM device.

reference	Semiconducting layer	Stretchability	Consumption (fJ)	V_{DS} (V)	Duration (ms)
[1]	P3HT-b-PPT(5F)/PMMA	Flexible	1.82	-1×10^{-3}	1000
[2]	Pbs QDs/PMMA/Pentacene	Flexible	0.55	-1×10^{-2}	100
[3]	Dif-TES-ADT	Flexible	0.07	-1×10^{-1}	250
[4]	C8-BTBT/P(VDT-TrFE)	Flexible	0.05	-2.5×10^{-5}	20
[5]	DPP-TT	Flexible	0.7×10^6	-2	200
[6]	PEDOT:PSS/GOPS	140% stretchable	1.98×10^{10} *	-0.6	132
[7]	SC-SWCTs+CdSe/ZnS QDs	80% stretchable	1.54×10^{-2}	-0.000001	20
[8]	CsPbBr ₃ QCM film /DPP-TT CONPHINE film	100% stretchable	0.015	-1×10^{-6}	100
[9]	N2200/s-CNT	50% stretchable	4000	1	50
[10]	P3HT NFs	50% stretchable	2.36×10^9 *	1	50
[11]	PTDPP-Se-6Si	50% stretchable	3.7×10^{-3} *	-1×10^{-4}	10
[12]	DPP-g2T	60% stretchable	1.995×10^8 *	-0.7	100
[13]	P3HT/PEO NWs	50% stretchable	1.975×10^{14} *	0.5	100

This work	DPP-TT/PBAT FMM film	125% stretchable	4.5×10 ⁶	-1	5000
-----------	----------------------	------------------	---------------------	----	------

The “*” in table indicated that the energy consumption were not published directly and were derived by the given equations and graphs.

Supplementary Table5. Comparison of the PPF index of reported stretchable synaptic phototransistors before and after stretching.

reference	Semiconducting layer	PPF index (%)	Δt (ms)	PPF index (%) at stretched states	$\frac{\Delta PPF}{PPF}$ (%)
[14]	IDT-BT	191	500	123 ($\epsilon=100\%$)	35.6
[8]	CsPbBr ₃ QCM film	148	500	232 ($\epsilon=100\%$)	56.7
[9]	N2200	177	50	157* ($\epsilon=50\%$)	-11.2
[9]	N2200/s-CNT	200	250	2.14 *($\epsilon=50\%$)	114
[9]	N2200/s-CNT	199	250	1.32 *($\epsilon=50\%$)	-33.6
[10]	P3HT NFs	146	40	114 *($\epsilon=50\%$)	-22
[11]	PTDPP-Se-6Si	185	1000	167 *($\epsilon=50\%$)	-10
[13]	P3HT/PEO NWs	273	50	146 *($\epsilon=50\%$)	-47
[6]	PEDOT:PSS/GOPS	477	132	380 *($\epsilon=100\%$)	-20
[7]	SC-SWCTs+CdSe/ZnS QDs	206	400	141 *($\epsilon=80\%$)	-32
This work	DPP-TT/PBAT FMM film	130	1000	138 ($\epsilon=100\%$)	6.7

The “*” in table indicated that the PPF index were not published directly and were derived by the given equations and graphs.

RESPONSES TO THE COMMENTS OF THE REVIEWER #3

Comment: In their manuscript, Zhou et al. introduced a stretchable semiconductor film featuring a geometry-deformable micromesh and nanofibril substructure, achieved through phase separation mediated by Van der Waals forces. This design, which combines the geometry-deformable micromesh and an interpenetrating phase within the blended semiconductor-poly(butylene adipate-co-terephthalate) (PBAT) film, enhances both the mechanical and electrical stretchability of the semiconductor. As a result, this fibrous stretchable semiconductor shows potential for applications in stretchable optoelectronic synaptic devices. This work is interesting, but there are

several issues need to be addressed.

Response: We are grateful for the reviewer's valuable comments and suggestions, which help to demonstrate the contribution of our strategy and improve the investigation on photophysical performance. We have revised the manuscript carefully according to each comment, as listed below.

1. This work needs more demonstration to show, how their approach significantly advances the field. Similar methodologies have been explored in existing literature, and the manuscript does not highlight distinct innovations or improvements.

Response: We appreciate the reviewer's valuable feedback regarding the novelty of our approach.

In order to highlight the distinct innovations and improvements of our work, we summarized the main innovations as follows:

(1) We applied the design concept "functional-orientated structure design" in the optimization of semiconducting polymer films towards their application in the emerging stretchable synaptic phototransistors. Guided by this concept, we constructed successfully highly stretchable and efficient semiconducting films with fibrous micromesh structure (FMM).

(2) Based on the phase separation strategy to tailor the film morphology, we emphasized the importance of intermolecular interactions in tuning the film structure. Meanwhile, we also revealed the dependence of strain dissipation mechanism on pore diameter, and identified a critical threshold of pore diameter to guide effective processing of mesh structure. Our work refined this strategy and is widely applicable for other hybrid systems.

(3) The obtained FMM film exhibited obviously improved performance compared to the film without modification or hybrid film employing SEBS as elastic component. To highlight the innovations, we also compared our work with reported results. Firstly, we compared the electrical performance of our FMM film with other blended organic semiconductor with mesh- or fibrous structure, emphasizing the structure dependent stability. A diagram explaining this statement is added in

Figure 5d, and we also add relevant statement in the manuscript in line 404-407.

Figure 5d. Comparison of mobility retention of stretchable OFETs between documented works and this work. All the OFETs were stretched to 100% strain.

“When stretched to 100% strain, N2200-based FMM film exhibited a high current retention of 200% relative to its initial state, significantly exceeding that of LS film, neat film and other hybrid films with porous structure or fibrous structure (Supplementary Fig. 30-33, Fig. 5d, Table 1)”

Meanwhile, we compared stretchable synaptic transistors in our work and reported works with respect to stability and mechanical stretchability. The corresponding diagram is added in **Figure 6f**. Compared to reported results, Our FMM film demonstrated the smallest deviation of PPF (6.7%) after stretching to 100%. Meanwhile, our FMM film exhibited unexpected two-fold improved light responsibility compared to the neat film. These results confirmed the benefit of FMM structure in application in stretchable synaptic transistors.

Figure 6f. Comparison of stretchability, PPF index and stability of PPF index achieved by reported stretchable organic synaptic transistors and our work. For each data point, the sphere radius was scaled by the deviation of PPF index ($r \propto \frac{\Delta PPF}{PPF0\%}$, ΔPPF is the

difference between PPF index under maximum strain and that of initial state).

2. Moreover, the experimental design lacks critical controls, making it difficult to validate the claims of improved performance. The demonstration of the device's synaptic performance is also weak, as it fails to maintain stable performance under deformation and exhibits only basic synaptic functionality.

Response: Thank you very much for your valuable comments.

In the revised manuscript, to demonstrate the improvement of the strategy, we compared the performance of FMM synaptic transistors with reported results, with respect to stretchability, PPF index and PPF index deviation under strain. Meanwhile, we compared the performance of synaptic transistors based on FMM film with neat film and LS film.

To the concern about the improvement of performance: On the one hand, compared with previously reported results, our FFM film based synaptic transistors exhibited stable PPF index under strain. PPF deviation of 6.7% was observed upon stretched to 100% strain, which is one of the most stable stretchable organic synaptic transistors (Supplementary Table 5, Figure 6f). Our PPF and EPSC exhibit greater stability under both stretching and cyclic stretching conditions. On the other hand, compared to the synaptic transistors employing neat film and LS film, the FMM synaptic transistors exhibited greater stability under both stretching and cyclic stretching conditions. In the revised manuscript, we discussed in Page 20 as below:

“Synaptic transistor of FMM film achieved a very stable PPF index, where only deviation of 6.7% was observed after stretching to 100% strain (Figure 6e). This is rather low compared to the neat-film based synaptic transistors and previously reported stretchable organic synaptic transistors. (Supplementary Fig. 42, Supplementary Table 5). Under increasing strain from 0% to 100%, the FMM synaptic transistors preserved 29% of the paired-pulse facilitated current, significantly outperforming the performance of LS and neat film devices.

...

To demonstrate cycle stability, synaptic phototransistors were characterized under 50%

strain for 10000 stretching-releasing cycles. FMM synaptic transistors exhibited stable device functionality at stretching state up to 1000 cycles and reversible performance at released state after 10000 cycles (Figure 6i, Supplementary Fig. 45). However, the LS synaptic transistor failed to exhibit light response after 1000 cycles.”

To the concern that the synaptic transistors only exhibited basic performance: In this work we demonstrated a strategy to obtain highly stretchable semiconducting polymer films with both improved stretchability and enhanced photo responsivity. As a proof of concept, we demonstrated the application of semiconductor films on synaptic phototransistors. Our demonstration of synaptic transistors exhibited capacity to imitate the visual memory of eyes and brain, including EPSC, PPF, STP and LTP. And the FMM synaptic transistors exhibited superior performance than synaptic transistor employing neat film and LS film. Although the demonstration is basic, we think it is capable to support the idea that semiconducting polymer films with FMM structure exhibit great potential for developing stretchable synaptic transistors.

3. The design of micromesh or nanoconfinement structures through precise control of phase separation in blend films of semiconductors and elastomers is well documented in previous works [e.g., Science, 2017, 355, 59; Nat. Electron. 2022, 5, 881]. The phase separation ratio can be effectively tuned by adjusting the ratio of the polymer semiconductor to elastomers. These methods have also been successfully employed in the development of stretchable optoelectronics [e.g., Chem. Eng. J., 2024, 492, 152143]. The authors claim they can precisely control the Van der Waals forces, but their method of tuning phase separation by changing the ratio of the semiconductor to PBAT to control fiber and pore diameters does not appear to differ from existing works [e.g., Science, 2017, 355, 59].

Response: We thank the reviewer for this observation and for highlighting key references.

To the concern about the difference between our strategy and the reference: While it is true that phase separation in blend systems has been employed for structural tuning, some challenges exist and impede its further application. In Table R2 we summarized

the results of reported researches based on similar methodologies, as a comparison of our work. All the films exhibit micromesh or nanoconfinement structures through precise control of phase separation. Generally, incorporating elastomer into the conjugated polymers improves the stretchability, owing to the mechanical compliance of elastomer/plastic polymers. Specifically, the mesh structure and fibrous structure contributes to the stretchability and light responsibility differently. For the purpose of improving light responsibility, fibrous structure is preferred owing to the abundant surface. When considering the mechanical stability, mesh structure is more promising, as the geometric deformation of pore structure dissipates part of the strain energy, as we discussion in section 2.3 in the manuscript. Thus, for the purpose of constructing semiconducting film serving in the stretchable synaptic phototransistors, which demand both high stretchability and high light responsivity, a promising film morphology is the combination of mesh structure and fibrils. However, this structure has not been reported until our work. Another challenge is the pronounced phase separation between the elastomer and conjugated polymers in the mesh structure, which leads to structure failure near the interface, resulting in performance degradation under larger strain. As shown in previous work (*Nat. Electron.* **2022**, 5, 881), LS films only retain their performance up to 50% strain. These challenges limit further application of “conjugated polymer-elastomer blending” strategy. Inspired by these works, our specific contributions lie in:

- (1) Compared to LS film employing commonly used elastomer SEBS, we achieved more interpenetrating dual network of semiconducting domain and stretchable domain in FMM films. The strategy provides a solution to the problem caused by pronounced phase separation.
- (2) We combined mesh structure and fibrous substructure in the blended films, which exhibited both superior stretchability and improved photo responsivity.
- (3) We introduced Van der Waals interaction of elastomer as a design parameter. This is a conceptual shift from conventional design strategy, which focuses on only ratio or inherent stretchability of elastomer. The impact of intermolecular interaction on the film morphology is discussed in Section 2.1 and 2.2 in the

manuscript.

(4) Meanwhile, we also revealed the dependence of strain dissipation mechanism on pore diameter, and identified a critical threshold of pore diameter to guide effective processing of mesh structure. This work fills the gap of this strategy and is widely applicable for other hybrid systems.

Table R2. Comparison of blended polymer semiconductors with respect to composition, film morphology, stretchability, and photo responsivity.

reference	OSC/elastomer	Ratio of elastomer	Film morphology	stretchability	$\frac{R^{hybrid}}{R^{neat}}$	$\frac{EPSC^{50\%}}{EPSC^{0\%}}$
[1]	DPPTT/SEBS	70 wt.%	confined fibrils	100%	no	no
[2]	N2200/PU	65 wt.%	micromesh	50%	no	no
[3]	DPPTT/SEBS	80%	Confined fibrils	50%	no	26.5% (//)* 60% (⊥)*
This work	DPPTT/PBAT	40%	Fibrous micromesh	100%	2	45.5% (⊥) 56% (//)

The “*” in table indicated that $\frac{EPSC^{50\%}}{EPSC^{0\%}}$ were not published directly and were derived by the given equations and graphs.

For the reviewer’s concern that we claimed that we can precisely control the Van der Waals forces. We are very sorry for the unclear expression in the original manuscript. We appreciate the reviewer for pointing this out and the opportunity to clarify our intended expression. In the manuscript (line 20-21 in the abstract section), our use of the sentence “Here, we fabricated biomimetic hybrid semiconducting films featuring geometry-deformable micromesh and nanofibril substructure, through the precise tuning of Van der Waals force-mediated phase-separation.” was not meant to imply that we are directly manipulating the Van der Waals forces themselves. Rather, we intended to convey that the phase separation process, which is mediated by intermolecular interactions including Van der Waals interactions, can be modulated. The fibrous mesh structure was obtained by incorporating PBAT, which is different from reported work. And the fibrils and pore diameters by adjusting the semiconductor-to-elastomer ratio,

which is the same as reported works (e.g., *Science*, **2017**, 355, 59).

To avoid misunderstanding, we have now revised the sentence in the manuscript (line 18-20) to read:

“Here, we fabricated biomimetic hybrid semiconducting films featuring geometry-deformable micromesh and nanofibril substructure, through the Van der Waals force-mediated phase-separation.”

We also added a clarification sentence in the Introduction (Page 2-3) to better distinguish our approach from prior studies.

“Notably, this strategy is capable to modulate the film morphology into structures resembling biological tissues, e.g., mesh structure²⁵ and fibrous textures^{26,27}, offering potential for enhancing bioinspired functionalities. However, current studies have yet to integrate these structural features to achieve optimal performance. Moreover, significant challenges remain, particularly in mitigating the effects of pronounced phase separation, which can disrupt the continuity of both charge and strain transport, leading to degradation under large deformation.”

4. The comparison between the FMM film (i.e., N2200-PBAT film) and the LS film (N2200-SEBS film) is unfair. The performance of the FMM film is optimized for the best PBAT/N2200 ratio, whereas the LS film, although using the same weight ratio as the FMM film, may not be at its optimal SEBS/N2200 ratio. The authors should also determine the optimal ratio for the LS film and then compare the performance improvements of their FMM film-based device to the best LS film-based device. Additionally, the authors should fabricate more devices using various polymer semiconductors and compare their performance. This will help verify that their method is broadly applicable and represents an advancement over existing techniques. Comparing only N2200-based FMM film with N2200-based LS film is not sufficiently convincing.

Response: Thank you very much for your meaningful comments.

About the question why we compare FMM film and LS film with the same composition ratio, which seems to be merely the optimized ratio of PBAT, we would

like to explain why we chose the ratio first. And for the concern that unfair comparison might exist, we investigated the mechanical stretchability, electrical performance of LS films with various compositions.

The ratio of hybrid film was chosen based on the stretchability, field effect charge carrier mobility, and the film morphology. For the mechanical stretchability, as shown in Figure 1e and Supplementary Fig.10, FMM films with 10 wt.%, 20 wt.% and 40 wt.% exhibited comparable crack onset strain (COS), indicating comparable mechanical strength. In term of electrical stretchability, although the inherent mobility of hybrid films decreases with the increasing ratio of PBAT, all the semiconducting films are eligible for organic electronics, demonstrating on-current within the range of 1 μA to 10 μA , and field effect mobility greater than $0.05 \text{ cm}^2 \text{ V}^{-1} \text{ s}^{-1}$. The third aspect is the morphology, i.e. the pore diameter of the porous hybrid film. Our study in Section 2.3 investigating the dependence of strain dissipation on pore diameter elucidates that porous film with higher pore diameter are more geometrically deformable, in which way the polymer chains will be less stretched, and prone to demonstrate better stability under strain. Therefore, considering all three aspects, we chose the hybrid film with the 40 wt.% for later study, with the anticipation of better electrical stretchability.

To demonstrate that this composition is also a good choice for LS films, we investigated the mechanical stretchability and inherent electrical performance of LS films with various compositions. As is shown below, the mechanical stretchability of LS increases with the ratio of SEBS. And the inherent field effect mobility decreases with the increasing ratio of SEBS. When the ratio of SEBS increases up to 40 wt.%, LS film exhibits mobility greater than $0.05 \text{ cm}^2 \text{ V}^{-1} \text{ s}^{-1}$.

Meanwhile, when the ratio of insulator is 40 wt.%, the averaged pore diameters of both films are greater than 700 nm (Supplementary Fig. 25). Thus, for both film the pore size is large enough for dissipating the strain energy under stretching.

Therefore, when applying the three criteria to LS film, the composition of 40 wt.% is also the optimizing choice.

Table R3. Electrical performance and crack-on-set strain of LS film with varying ratio of SEBS.

PBAT Contents	μ_{sat} ($\text{cm}^2\text{V}^{-1}\text{s}^{-1}$)	V_{th} (V)	I_{on}/I_{off}	I_{off} (nA)	Crack-onset strain (%)
0%	0.11	13.7	2.6×10^3	11.3	50
10%	0.14	11.8	4.6×10^3	8.3	110
20%	0.094	16.9	8.1×10^3	4.0	140
40%	0.075	14.9	4.1×10^3	6.8	150

To demonstrate the universality of our FMM strategy, we also applied this strategy to another semiconducting polymer DPP-TT. The morphology and electrical performance of DPPTT-based hybrid film were investigated. For comparison, the neat DPPTT film and LS film comprising DPPTT and SEBS were also prepared. Similar with the morphology in N2200-based hybrid films, DPPTT-based hybrid film comprising PBAT exhibited a desired fibrous micromesh structure, which contributed to the aggregation effect during phase separation. While the LS film of DPPTT/SEBS exhibited an ordinary mesh structure with larger pore diameters, resulting from the weak phase separation effect (Supplementary Fig. 32). Electrical performance of stretchable OFETs employing the DPPTT-based neat film, LS film and FMM film were investigated under tensile strain from 0% to 100% (Supplementary Fig. 33). Compared to neat film and LS film, FMM film exhibited higher electrical stability, especially when large strain was applied. These results demonstrated the versatility of our strategy in conjugated polymers and reconfirmed the beneficial effect of FMM structure on electrical stretchability.

5. The demonstrated synaptic phototransistor performance is not very promising, as similar results have already been widely reported in the literature. Additionally, the strain significantly affects the device's synaptic performance, making it challenging to use in practical applications. Moreover, the comparison of synaptic performance is only

between the neat film and the FMM film. Why not include the synaptic performance of the LS film? Is the synaptic performance of the FMM film superior to that of the LS film?

Response: Thank you very much for your comments.

To demonstrate the performance of our synaptic transistors, we compared the device parameters of our synaptic transistors with results reported in other works in the revised supplementary information (Supplementary Table 4, Supplementary Table 5), focusing on the energy consumption, PPF index and stability of PPF index. Our synaptic transistors exhibited very good stability under large strain and moderate PPF index among synaptic transistors employing organic semiconductors. Only a deviation of 7% was observed in the FMM-synaptic transistor upon stretching to 100%, which is much more stable than other reported stretchable synaptic transistors (Figure 6f). While the energy consumption of our synaptic transistors is about 10^6 times that of other synaptic transistors employing inorganic semiconductors. The high energy consumption is a common issue in the synaptic transistors which only utilize organic semiconductors, which is attributed to the relative low charge carrier mobility in the organic semiconductors. But this problem can be well resolved, for example via mixing quantum particle of inorganic semiconductors in the conjugated polymer films. Thus, although our synaptic transistors do not exhibit excellent performance in all aspects, they exhibit basic capacity to imitate the synapses and also exhibit high stability under strain, which are capable to show the advantage of the FMM structure.

Meanwhile, we appreciate your suggestion that compare the performance of LS-film based synaptic transistors with the FMM film. In the revised manuscript, we added the photo responsivity of LS film in Figure 6f and Supplementary Fig. 39, performance of the LS-synaptic transistors under strain from 0% to 100% (Supplementary Fig. 42), as well as cyclic performance under strain of 50% (Supplementary Fig. 45). Compared to FMM film, LS film exhibited reduced photoresponsibility, equivalent stability of PPF index under strain and much less EPSC under cyclic stretching. The performance

of neat film, FMM film and LS film of DPP-TT in synaptic transistors were discussed in the revised manuscript (line 506-510, 527-532 line) as follows:

“Meanwhile, synaptic phototransistors employing FMM films exhibit a twofold photoresponsivity compared to the neat film-based phototransistors (Figure 6e, Supplementary Fig. 40). On the contrast, LS film exhibited reduced photo sensitivity than the neat film. These results indicate that FMM structures enhance the light harvesting process via the wide-angle light absorption of nanofibers.”

“To demonstrate the cycle stability of devices, synaptic phototransistors were characterized under 50% strain for 10000 stretching-releasing cycles. FMM synaptic transistors exhibited stable device functionality at stretching state up to 1000 cycles and reversible performance at released state after 10000 cycles (Fig. 6i, Supplementary Fig. 45). However, the LS synaptic transistor failed to exhibit light response after 1000 cycles.”

Figure 6f. comparison of stretchability, PPF index and stability of PPF index achieved by reported stretchable organic synaptic transistors and our work. For each data point, the sphere radius was scaled by the deviation of PPF index ($r \propto \frac{\Delta PPF}{PPF^{0\%}}$, ΔPPF is the difference between PPF index under maximum strain and that of initial state).

Supplementary Fig. 40. Photoelectric responses of the synaptic transistors employing (a) FMM and (b) neat film and (c) LS film of DPP-TT under varying irradiation intensities.

Supplementary Fig. 42. EPSC under paired pulses of the stretchable synaptic phototransistors under strain. (a) Synaptic transistor employing neat film, (b) Synaptic transistor employing FMM film, (c) Synaptic transistor employing LS film. (d) PPF index of the stretchable synaptic transistors under strain. The irradiation power was 10 mW cm^{-2} . The wavelength was 760 nm .

Supplementary Fig. 45. Photoelectronic stability of the synaptic phototransistors employing FMM film and LS film of DPP-TT upon repeating stretching-releasing cycles (up to 10000 cycles) at 50% strain. The irradiation intensity was 1 mW cm^{-2} .

6. The change in the PPF index with pulse time interval is unusual. As shown in Figure 6c, the value initially shows a very fast decrease, and then slightly increases as the time interval increases. Why is that?

Response: Thank you very much for your comments.

The pair-pulse facilitation (PPF) in synaptic transistors are derived from the trapped excitons, which release charge carriers gradually after irradiation. These charge carriers contribute to light current in the subsequent irradiation, resulting in an improved light current compared to the former irradiation. Due to the life time of trapped charge carriers, the PPF index shows fast decrease with the interval increases. And with the interval increase to long period, e.g. 60s, the PPF effect becomes weak and the EPSC is modulated by other complexed mechanism, such as long-term memory. In this phase, the random system noise becomes pronounced. That is why the PPF index demonstrates minor variation when the time interval becomes large, such as 40 s and

60 s. That is to say, in these cases, PPF is not appropriate to describe the light induced current. Therefore, in the revised manuscript, we deleted the data of 60 s.

Meanwhile, we also fitted the PPF index data with double-exponential curve. As shown in Figure 6c, the experimental results were fitted as

$$PPF = C_0 + C_1 e^{-\frac{t}{t_1}} + C_2 e^{-\frac{t}{t_2}}$$

where t_1 is 3.85, t_2 is 631.4. t_1 and t_2 represent the rapid decay relaxation and the slow relaxation of the EPSC.

7. In Figure 6d, the final device current state after 1 second of irradiation is larger than that after 2 seconds and 5 seconds of light irradiation. Could you explain the reason for this? It appears that the device state after light irradiation does not fully return to the initial state. Can the memory states of this device be erased?

Response: Thank you very much for your careful comments.

We noticed that the final device current state after 1 second of irradiation is larger than that after 2 seconds and 5 seconds of light irradiation. According to our observations, the reason is that during the measurement, the dark current (base line) has not reached the state and was still increasing slightly. After several seconds, the base line increased slightly, which resulted in a current difference between the dark current before irradiation and after decay. Thus, to avoid misunderstanding, we remeasured the data and updated Figure 6d.

Figure 6d. EPSC of FMM film-based synapse under different irradiation durations.

To verify whether the system returns to its original state after irradiation of 20s, we measured the EPSC after irradiation for over 200s, and according to the results, the

EPSC will return to the original state after 200 s. We added the information in the revised manuscript line 505-506 (Page 21) and in the Supplementary Fig.40.

In line 505-506:

“Notably, this nonvolatile memory gradually decays to the original state after 200 s (Supplementary Fig. 39).”

To the question that “Can the memory states of this device be erased?”: We tried to erase the light current via gate voltage (constant voltage and pulse). As shown in Figure R3, when a positive bias was applied, the current in the channel decrease instantly. But current went back to the original state instantly after the gate bias was removed. Thus, the memory states of this device can not be erased.

Figure R3. Drain current of FMM synaptic transistors under various conditions of gate bias after light stimulus. (a) Drain current under single pulse of gate bias ($V_G=60$ V) after irradiation. The width of electrical pulses were 2 s and 5 s, respectively. (b) Drain current under multiple continuous pulses of gate bias ($V_G=60$ V) after irradiation. The width of electrical pulses was 2 s.

8. How stable is the device under long-term light irradiation? How many cycles can the device operate during repeated light on and off conditions? Additionally, how consistent is the EPSC value across different test cycles for the same device, and how repeatable is it from one device to another?

Response: Thank you for your insightful comments.

To evaluate the performance consistency of FMM synaptic transistors, we also studied the cycling endurance under repeating light stimulus and device to device repeatability.

To demonstrate the cyclic endurance, device was subjected to continuous light on and off cycles (width is 1s and power is 1 mW/cm^2) for more than 350 s. The EPSC was summarized as follows (Figure R4 and Table R4). According to the results, EPSC declined under repeating irradiation. After 150 cycles, EPSC decreased from 19.25 nA to 2.3 nA. The PPF index also declines with the irradiation time prolongs. PPF index decreased to 1.04 after 100 cycles and increased to 1.32 again after 300 cycles. This could be related to complex interactions between the traps, charge carrier and performance stability of the laser source.

Figure R4. EPSC of the FMM synaptic transistor under repeating light on and off cycles. The pulse width is 1s and the power is 1 mW/cm^2 .

Table R4. Cyclic endurance of FMM synaptic transistors under repeating light on and off cycles. $t=100 \text{ s}$.

Sample	A1 (nA)	A5 (nA)	A5/A1
0	19.25	65.79	3.42
1t	8.64	20.2	2.33
2t	3.88	4.04	1.04

3t	2.3	3.03	1.32
3.5t	2.6	3.58	1.38

To demonstrate the device to device repeatability, 5 devices were measured in 3 individual tests under irradiation of 1 mW/cm², pulse width was 1s. The EPSC were summarized in Table R5. For single device, the fluctuation of EPSC is between 20% ~ 65% of the averaged EPSC of devices. And the fluctuation from device to devices is 43% of the averaged EPSC of all the investigated devices. The fluctuation should be attributed to the film quality, traps in the semiconducting materials and limited cyclic endurance of the film.

Table R5. EPSC of five FMM synaptic transistors in 3 individual tests.

Sample	EPSC1 (nA)	EPSC2 (nA)	EPSC3 (nA)	Mean (nA)	STD
Device 1	9.8	8.17	8.05	8.67	0.98
Device 2	13.7	9.8	7.09	10.2	3.32
Device 3	9.65	7.71	5.32	7.56	2.17
Device 4	13.23	11.52	10.21	11.65	1.51
Device 5	11.95	9.02	7.18	9.38	2.40

9. The crack onset strain (COS) test depicted in Figure 1e should include error bars to enhance the accuracy of the results. Additionally, it is important to specify the substrate used, as it can significantly impact the COS value.

Response: We appreciate the reviewer’s thoughtful suggestion.

In the revised manuscript, we have added error bars to Figure 1e to reflect the standard deviation obtained from five independent measurements, which improves the clarity and reliability of the COS data. The raw data is displayed in this response (Figure R5-R8). Furthermore, we have specified the substrate used during the COS test in the figure caption of Figure 1e (Line 170-173) and the main text (Line 145-146), as it indeed plays an important role in influencing the mechanical behavior of the film.

PDMS substrate with a thin layer of SEBS was used as the substrate. The thickness of PDMS substrate and SEBS were kept as the same as the stretchable OFETs.

Figure R5. Optical microscope images of the neat film under strain. The COS test was conducted on the PDMS substrate with a thin layer of SEBS. The preparation of PDMS substrate and SEBS layer is detailed in the Method section. The scale bar is 20 μm .

Figure R6. Optical microscope images of the hybrid polymer film with the insulator (PBAT) contents of 10% under strain. The COS test was conducted on the PDMS substrate with a thin layer of SEBS. The preparation of PDMS substrate and SEBS layer is detailed in the Method section. The scale bar is 20 μm .

Figure R7. Optical microscope images of the hybrid polymer film with the insulator (PBAT) contents of 20% under strain. The COS test was conducted on the PDMS substrate with a thin layer of SEBS. The preparation of PDMS substrate and SEBS layer is detailed in the Method section. The scale bar is 20 μm .

Figure R8. Optical microscope images of the hybrid polymer film with the insulator (PBAT) contents of 40% under strain. The COS test was conducted on the PDMS substrate with a thin layer of SEBS. The preparation of PDMS substrate and SEBS layer is detailed in the Method section. The scale bar is 20 μm .

10. The stretching cyclic stability of the FMM-based device should be compared to that of the LM-based device.

Response: Thank you for your valuable comments.

In the revised manuscript, we compared the cyclic stability of FMM film with LS film. The synaptic phototransistors employing FMM film and LS film of DPP-TT upon repeating stretching-releasing cycles (up to 10000 cycles) at 50% strain. Corresponding

EPSC were summarized in Supplementary Fig. 45. We discussed the cyclic stability of devices in the manuscript (line 527-532), as listed below:

Supplementary Fig. 45. Photoelectronic stability of the synaptic phototransistors employing FMM film and LS film of DPP-TT upon repeating stretching-releasing cycles (up to 10000 cycles) at 50% strain. The irradiation intensity was 1 mW/cm^2 .

Line 527-532 in revised manuscript:

“To demonstrate the cycle stability of devices, synaptic phototransistors were characterized under 50% strain for 10000 stretching-releasing cycles. FMM synaptic transistors exhibited stable device functionality at stretching state up to 1000 cycles and reversible performance at released state after 10000 cycles (Fig. 6i, Supplementary Fig. 45). However, the LS synaptic transistor failed to exhibit light response after 1000 cycles.”

11. The film's stability has not been adequately addressed. It is essential to evaluate whether the film morphology will change over prolonged storage durations. According to a study published in Science [Science, 2017, 355, 59], nanoconfinement films have demonstrated the capability to sustain stable device performance when stored for up to one year. Therefore, the authors are urged to explore the stability of both their film and devices over extended storage periods and under varying humidity and temperature

conditions.

Response: We thank the reviewer for this insightful comment. We agree that the long-term stability of the film and devices is an important consideration. In response, we have performed additional storage experiments to evaluate the morphological and performance stability of our films and devices over an extended period. Specifically, samples were stored in ambient conditions (approximately 25 °C and 46% relative humidity) for 30 days, after which we re-evaluated film morphology using AFM and optical microscope, and measured device performance of rigid devices and stretchable devices. The results indicate minimal changes in film morphology either and only a marginal decline in device performance when storage in nitrogen, suggesting good stability of this FMM film. These mentioned data have been included in the revised manuscript (see Supplementary Fig. 34 and line 416-424), and also attached here:

Supplementary Fig.34. Stability of film morphology and devices performance stored in glove box and humidity control cabinet. The adhesion images of FMM films of N2200 a) as annealed, b) stored in nitrogen for 30 days c) stored in air for 30 days. The optical microscope images of FMM films of N2200 under strain of 50% at different storage conditions d) the original film, e) in nitrogen after 10 days and f) in air after 10 days. The stretching direction is displayed by the arrows inserted. Transfer curves of OFETs on Si/SiO₂ based on FMM films of N2200 at different storage conditions g) in nitrogen and h) in air. (i) The corresponding mobility of devices in (g) and (h). Transfer curves of stretchable OFETs based on FMM films of N2200 at different storage conditions j) in nitrogen and k) in air. (l) Corresponding mobility of devices in (j) and

(k). The temperature and humidity in glove box filled with nitrogen and humidity control cabinet is 21 °C and 16 ppm and 25 °C and 46% RH.

line 435-441 in the revised manuscript

“Meanwhile, we also investigated the stability of film morphology and device performance under different temperature and humidity. As shown in Supplementary Fig. 34, the fibrous mesh structure of FMM films of N2200 exhibited minimal changes after being stored at nitrogen or ambient condition for one month. Devices on Si/SiO₂ substrates also exhibited stable performance, where high mobility retention of 83% and 59% were obtained in the devices stored in nitrogen and humidity control cabinet, respectively. The stretchable OFETs also demonstrated stable performance under strain of 50% after 30-day storage in nitrogen and ambient condition (46% RH).”

12. The use of the term "biomimetic" in the title may be misleading. While I acknowledge that the film exhibits an abundance of fiber-like structures, it is challenging to draw a direct connection between these structures and the complex photoreceptor cells found in the eyes, as there is no functional similarity. Additionally, the photoresponse primarily originates from the semiconductor itself rather than from the fiber-like structure design.

Response: We thank the reviewer for the thoughtful comment.

By using the term “biomimetic”, our intent was to emphasize the structural resemblance of the fiber-like morphology in our films to the photoreceptor cells in the retina. We are sorry that in the original manuscript the idea was not expressed properly. In the revised manuscript we drew an illustration to demonstrate the structural similarity between the photoreceptor cells and the fibrous semiconducting layers.

Meanwhile, we do agree with the idea that the photo-responsibility is from the semiconducting layer itself, not from the fibrous structure. But as we stated in the manuscript, wide-angle light absorption of nanofibers in FMM films facilitate the photo-responsibility of the semiconducting layers. Such fibrous structure is also founded in the photoreceptor layer, which is thought to enhance the responsivity of the

retina.

The functional similarity is that both the semiconducting layers and photoceptor cells trigger current upon the irradiation. And the structural similarity is that both the FMM film and the photoreceptor cells share the fiber-like structure, which is supposed to facilitate the harvest of light.

Reference

1. Xu, J. *et al.* Highly stretchable polymer semiconductor films through the nanoconfinement effect. *Science* **355**, 59–64 (2017).
2. Guan, Y.-S. *et al.* Elastic electronics based on micromesh-structured rubbery semiconductor films. *Nature Electronics* **5**, 881–892 (2022).
3. Nam, T. U. *et al.* Intrinsically stretchable phototransistors with polymer-QD-polymer multi-layered hybrid films for visible-NIR perspective electronic skin sensors. *Chemical Engineering Journal* **492**, 152143 (2024).